# PID-Inspired Inductive Biases for Deep Reinforcement Learning in Partially Observable Control Tasks

**Ian Char**
Machine Learning Department
Carnegie Mellon University
Pittsburgh, PA 15213
ichar@cs.cmu.edu

**Jeff Schneider**
Machine Learning Department, Robotics Institute
Carnegie Mellon University
Pittsburgh, PA 15213
schneide@cs.cmu.edu

## Abstract

Deep reinforcement learning (RL) has shown immense potential for learning to control systems through data alone. However, one challenge deep RL faces is that the full state of the system is often not observable. When this is the case, the policy needs to leverage the history of observations to infer the current state. At the same time, differences between the training and testing environments makes it critical for the policy not to overfit to the sequence of observations it sees at training time. As such, there is an important balancing act between having the history encoder be flexible enough to extract relevant information, yet be robust to changes in the environment. To strike this balance, we look to the PID controller for inspiration. We assert the PID controller's success shows that only summing and differencing are needed to accumulate information over time for many control tasks. Following this principle, we propose two architectures for encoding history: one that directly uses PID features and another that extends these core ideas and can be used in arbitrary control tasks. When compared with prior approaches, our encoders produce policies that are often more robust and achieve better performance on a variety of tracking tasks. Going beyond tracking tasks, our policies achieve 1.7x better performance on average over previous state-of-the-art methods on a suite of locomotion control tasks. [1]

## 1 Introduction

Deep reinforcement learning (RL) holds great potential for solving complex tasks through data alone, and there have already been exciting applications of RL in video game playing [70], language model tuning [53], and robotic control [2]. Despite these successes, there still remain significant challenges in controlling real-world systems that stand in the way of realizing RL's full potential [20]. One major hurdle is the issue of partial observability, resulting in a Partially Observable Markov Decision Process (POMDP). In this case, the true state of the system is unknown and the policy must leverage its history of observations. Another hurdle stems from the fact that policies are often trained in an imperfect simulator, which is likely different from the true environment. Combining these two challenges necessitates striking a balance between extracting useful information from the history and avoiding overfitting to modelling error. Therefore, introducing the right inductive biases to the training procedure is crucial.

The use of recurrent network architectures in deep RL for POMDPs was one of the initial proposed solutions [29] and remains a prominent approach for control tasks [47, 75, 50]. Theses architectures are certainly flexible; however, it is unclear whether they are the best choice for control tasks,

---

[1]Code available at https://github.com/IanChar/GPIDE

37th Conference on Neural Information Processing Systems (NeurIPS 2023).

especially since they were originally designed with other applications in mind such as natural language processing.

In contrast with deep RL methods, the Proportional-Integral-Derivative (PID) controller remains a cornerstone of modern control systems despite its simplicity and the fact it is over 100 years old [5, 48]. PID controllers are single-input single-output (SISO) feedback controllers designed for tracking problems, where the goal is to maintain a signal at a given reference value. The controller adjusts a single actuator based on the weighted sum of three terms: the current error between the signal and its reference, the integral of this error over time, and the temporal derivative of this error. PID controllers are far simpler than recurrent architectures and yet are still able to perform well in SISO tracking problems despite having no model for the system's dynamics. We assert that PID's success teaches us that in many cases only two operations are needed for successful control: summing and differencing.

To investigate this assertion, we conduct experiments on a variety of SISO and multi-input multi-output (MIMO) tracking problems using the same featurizations as a PID controller to encode history. We find that this encoding often achieves superior performance and is significantly more resilient to changes in the dynamics during test time. The biggest shortcoming with this method, however, is that it can only be used for tracking problems. As such, we propose an architecture that is built on the same principles as the PID controller, but is general enough to be applied to arbitrary control problems. Not only does this architecture exhibit similar robustness benefits, but policies trained with it achieve an average of *1.7x better performance* than previous state-of-the-art methods on a suite of locomotion control tasks.

## 2 Preliminaries

**The MDP and POMDP**   We define the discrete time, infinite horizon Markov Decision Process (MDP) to be the tuple $(\mathcal{S}, \mathcal{A}, r, T, T_0, \gamma)$, where $\mathcal{S}$ is the state space, $\mathcal{A}$ is the action space, $r : \mathcal{S} \times \mathcal{A} \times \mathcal{S} \to \mathbb{R}$ is the reward function, $T : \mathcal{S} \times \mathcal{A} \to \Delta(\mathcal{S})$ is the transition function, $T_0 \subset \Delta(\mathcal{S})$ is the initial state distribution, and $\gamma$ is the discount factor. We use $\Delta(\mathcal{S})$ to denote the space of distributions over $\mathcal{S}$. Importantly, the Markov property holds for the transition function, i.e. the distribution over a next state $s'$ depends only on the current state, $s$, and current action, $a$. Knowing previous states and actions does not provide any more information. The objective is to learn a policy $\pi : \mathcal{S} \to \Delta(\mathcal{A})$ that maximizes the objective $J(\pi) = \mathbb{E}\left[\sum_{t=0}^{\infty} \gamma^t r(s_t, a_t, s_{t+1})\right]$, where $s_0 \sim T_0$, $a_t \sim \pi(s_t)$, and $s_{t+1} \sim T(s_t, a_t)$. When learning a policy, it is often key to learn a corresponding value function, $Q^\pi : \mathcal{S} \times \mathcal{A} \to \mathbb{R}$, which outputs the expected discounted returns after playing action $a$ at state $s$ and then following $\pi$ afterwards.

In a Partially Observable Markov Decision Process (POMDP), the observations that the policy receives are not the true states of the process. In control this may happen for a variety of reasons such as noisy observations made by sensors, but in this work we specifically focus on the case where aspects of the state space remain unmeasured. In any case, the POMDP is defined as the tuple $(\mathcal{S}, \mathcal{A}, r, T, T_0, \Omega, \mathcal{O}, \gamma)$, where $\Omega$ is the space of possible observations, $\mathcal{O} : \mathcal{S} \times \mathcal{A} \to \Delta(\Omega)$ is the conditional distribution of seeing an observation, and the rest of the elements of the tuple remain the same as before. The objective remains the same as the MDP, but now the policy and value functions are not allowed access to the state.

Crucially, the Markov property does not hold for observations in the POMDP. That is, where $o_{1:t+1} := o_1, o_2, \ldots, o_{t+1}$ are observations seen at times 1 through $t + 1$, $o_{1:t-1} \not\perp o_{t+1} | o_t, a_t$. A naive solution to this problem is to instead have the policy take in the history of the episode so far. Of course, it is usually infeasible to learn a policy that takes in the entire history for long episodes since the space of possible histories grows exponentially with the length of the episode. Instead, one can encode the information into a more compact representation. In particular, one can use an encoder $\phi$ which outputs an encoding $z_t = \phi(o_{1:t}, a_{1:t-1}, r_{1:t-1})$ (note that encoders need not always take in the actions and rewards). Then, the policy and Q-value functions are augmented to take in $(o_t, z_t)$ and $(o_t, a_t, z_t)$, respectively.

**Tracking Problems and PID Controllers.**   We first focus on the tracking problem, in which there are a set of signals that we wish to maintain at given reference values. For example, in espresso machines the temperature of the boiler (i.e. the signal) must be maintained at a constant reference temperature, and a controller is used to vary the boiler's on-off time so the temperature

is maintained at that value [43]. Casting tracking problems as discrete time POMDPs, we let $o_t = \left( x_t^{(1)}, \ldots, x_t^{(M)}, \sigma_t^{(1)}, \ldots, \sigma_t^{(M)} \right)$ be the observation at time $t$, where $x_t^{(i)}$ and $\sigma_t^{(i)}$ are the $i^{\text{th}}$ signal and corresponding reference value, respectively. The reward at time $t$ is simply the negative error summed across dimensions, i.e. $-\sum_{m=1}^{M} \left| x_t^{(m)} - \sigma_t^{(m)} \right|$.

When dealing with a single-input single-output (SISO) system (with one signal and one actuator that influences the signal), one often uses a Proportional-Integral-Derivative (PID) controller: a feedback controller that is often paired with feedforward control. This controller requires no knowledge of the dynamics, and simply sets the action via a linear combination of three terms: the error (P), the integral of the error (I), and the derivative of the error (D). When comparing other architectures to the PID controller, we will use orange colored text and blue colored text to highlight similarities between the I and D terms, respectively. Concretely, the policy corresponding to a discrete-time PID controller is defined as

$$
\pi^{\text{PID}}(o_t) = K_P(x_t^{(1)} - \sigma_t^{(1)}) + K_I \sum_{i=1}^{t} (x_i^{(1)} - \sigma_i^{(1)}) dt + K_D \frac{\left( x_t^{(1)} - \sigma_t^{(1)} \right) - \left( x_{t-1}^{(1)} - \sigma_{t-1}^{(1)} \right)}{dt}
\tag{1}
$$

where $K_P$, $K_I$, and $K_D$ are scalar values known as gains that must be tuned. PID controllers are designed for SISO control problems, but many real-world systems are multi-input multi-output (MIMO). In the case of MIMO tracking problems, where there are $M$ signals with $M$ corresponding actuators, one can control the system with $M$ separate PID controllers. However, this assumes there is a clear breakdown of which actuator influences which signal. Additionally, there are often interactions between the different signals, which the PID controllers do not account for. Beyond tracking problems, it is less clear how to use PID controllers without substantial engineering efforts.

## 3 Methodology

To motivate the following, consider the task of controlling a tokamak: a toroidal device that magnetically confines plasma and is used for nuclear fusion. Nuclear fusion holds the promise of providing an energy source with few drawbacks and an abundant fuel source. As such, there has recently been a surge of interest in applying machine learning [1, 13], and especially RL [14, 45, 17, 71, 65, 64, 44], for tokamak control. However, applying deep RL has the same problems as mentioned earlier; the state is partially observable since there are aspects of the plasma's state that cannot be measured in real time, and the policy must be trained before-hand on an imperfect simulator since operation of the actual device is extremely expensive.

How should one choose a historical encoder with these challenges in mind? Previous works [50, 46] suggest using Long Short Term Memory (LSTM) [31], Gated Recurrent Units [15], or transformers [69]. These architectures have been shown to be powerful tools in natural language processing, where there exist complicated relationships between words and how they are positioned with respect to each other. However, do the same complex temporal relationships exist in something like tokamak control? The fact that PID controllers have been successfully applied for feedback control on tokamaks suggests this may not be the case [72, 27]. In reality, the extra flexibility of these architectures may become a hindrance when deployed on the physical device if they overfit to quirks in the simulator.

In this section, we present two historical encoders that we believe have good inductive biases for control. They are inspired by the PID controller in that they only sum and difference in order to combine information throughout time. Following this, in Section 5, we empirically show the benefits of these encoders on a number of control tasks including tokamak control.

**The PID Encoder.** Under the framework of a policy that uses a history encoder, the standard PID controller (1) is simply a linear policy with an encoder that outputs the tracking error, the integral of the tracking error, and the derivative of the tracking error. This notion can be extended to MIMO problems and arbitrary policy classes, resulting in the *PID-Encoder* (PIDE). Given input $o_{1:t}$, this encoder outputs a $3M$ dimensional vector consisting of $\left( x_t^{(m)} - \sigma_t^{(m)} \right)$, $\sum_{i=1}^{t} (x_i^{(m)} - \sigma_i^{(m)}) dt$, and $\frac{\left( x_t^{(m)} - \sigma_t^{(m)} \right) - \left( x_{t-1}^{(m)} - \sigma_{t-1}^{(m)} \right)}{dt}$ $\forall m = 1, \ldots, M$. For SISO problems, policies with this encoder have access to the same information as a PID controller. However, for MIMO problems the policy has

access to all the information that each PID controller, acting in isolation, would have. Ideally a sophisticated policy would coordinate each actuator setting well.

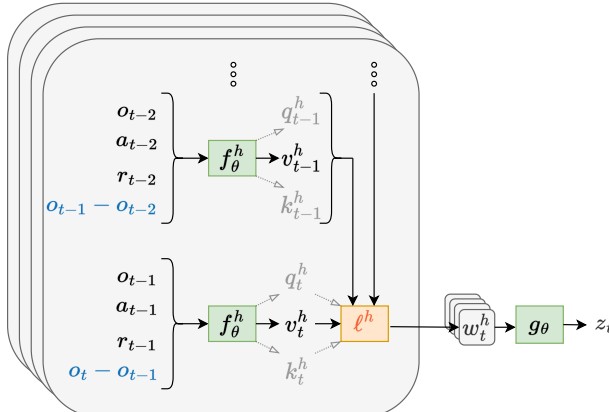

Figure 1: **Architecture for GPIDE.** The diagram shows how one encoding, $z_t$, is formed. Each of the gray, rounded boxes corresponds to one of the heads that makes up GPIDE. Each green box shows a function to be learned from data, and the orange box shows the weighted summation of all previous vectors, $v_{1:t}^h$. We write the difference in observations in blue text to highlight the part of GPIDE that relates to a PID controller's D term. Note that $q_{1:t}^h$ and $k_{1:t}^h$ only play a role in this process if head $h$ uses attention; as such, we write these terms in gray text.

**The Generalized PID Encoder.** A shortcoming of PIDE is that it is only applicable to tracking problems since it operates over tracking error explicitly. A more general encoder should instead accumulate information over arbitrary features of each observation. With this in mind, we introduce the *Generalized-PID-Encoder* (GPIDE).

GPIDE consists of a number of "heads", each accumulating information about the history in a different manner. When there are $H$ heads, GPIDE forms history encoding, $z_t$, through the following:

$$v_i^h = f_\theta^h(\text{concatenate}(o_{i-1}, a_{i-1}, r_{i-1}, o_i - o_{i-1})) \qquad \forall i \in \{1, \ldots, t\}, h \in \{1 \ldots, H\}$$
$$w_t^h = \ell^h(v_{1:t}^h) \qquad\qquad\qquad\qquad\qquad\qquad\qquad \forall h \in \{1 \ldots, H\}$$
$$z_t = g_\theta(\text{concatenate}(w_t^1, w_t^2, \ldots, w_t^h))$$

Here, GPIDE is parameterized by $\theta$. For head $h$, $f_\theta^h$ is a linear projection of the previous observation, action, reward, and difference between the current and previous observation to $\mathbb{R}^D$, and $\ell^h$ is a weighted summation of these projections. $g_\theta$ is a decoder which combines all of the information from the heads. A diagram of this process is shown in Figure 1. Note that $\theta$ is trained along with the policy and Q networks end-to-end with a gradient based optimizer.

Notice that the key aspects of the PID controller are present here. The difference in observations is explicitly taken before the linear projection $f_\theta^h$. We found that this simple method works best for representing differences when the observations are scalar descriptions of the state (e.g. joint positions). Although we do not consider image observations in this work, we imagine a similar technique could be done by taking the differences in image encodings. Like the integral term of the PID, $\ell^h$ also accumulates information over time. In the following, we consider several possibilities for $\ell^h$, and we will refer to these different choices as "head types" throughout this work. We omit the superscript $h$ below for notational convenience.

**Summation.** Most in line with PID, the projections can be summed, i.e. $\ell(v_{1:t}) = \sum_{i=1}^t v_i$.

**Exponential Smoothing.** In order to weight recent observations more heavily, exponential smoothing can be used. That is, $\ell(v_{1:t}) = (1 - \alpha)^{t-1} v_1 + \sum_{i=2}^t \alpha(1 - \alpha)^{t-i} v_i$, where $0 \leq \alpha \leq 1$ is the smoothing parameter. Unlike summation, this head type cannot accumulate information in the same way because it is a convex combination.

**Attention.** Instead of hard-coding a weighted summation of the projections, this weighting can be learned through attention [69]. Attention is one of the key components of transformers because of its ability to learn relationships between tokens. To implement this, two additional linear functions should be learned that project concatenate$(o_{i-1}, a_{i-1}, r_{i-1}, o_i - o_{i-1})$ to $\mathbb{R}^D$. These new projections are referred to as they key and query vectors, denoted as $k_i$ and $q_i$ respectively. The softmax between their inner products is then used to form the weighting scheme for $v_{1:t}$. We can rewrite the first two

steps of GPIDE for a head that uses attention as

$$v_i, k_i, q_i = f_\theta(\text{concatenate}(o_{i-1}, a_{i-1}, r_{i-1}, o_i - o_{i-1})) \quad \forall i \in \{1, \ldots, t\}$$

$$w_{1:t} = \ell(q_{1:t}, k_{1:t}, v_{1:t}) = \text{softmax}\left(\frac{q_{1:t}k_{1:t}^T}{\sqrt{D}}\right)v_{1:t}$$

Here, $q_{1:t}$, $k_{1:t}$, and $v_{1:t}$ are treated as $t \times D$ dimensional matrices. Since it results in a convex combination, attention has the capacity to reproduce exponential smoothing but not summation.

To anchor the GPIDE architecture back to the PID controller, we note that the P, I, and D terms can be formed exactly. At a high level, this is achieved when $f_\theta^h$ simply subtracts the target from the state measurement and when using summation and exponential smoothing heads with $\alpha = 1$. We write down the specific instance of GPIDE that results in these terms in Appendix A.1. While it is trivial for GPIDE to reconstruct the P, I, and D terms, it is less clear how an LSTM or GRU would achieve this, especially because of the I term. At the same time, GPIDE is much more flexible than the PID-representation since altering $f_\theta^h$ results in different representations at each time step and altering the type of head results in different temporal relationships.

## 4   Related Work

A control task may be partially observable for a myriad of reasons including unmeasured state variables [28, 75, 29], sensor noise[47], and unmeasured system parameters [76, 54]. When there are unmeasured system parameters, this is usually framed as a meta-reinforcement learning (MetaRL) [73] problem. This is a specific subclass of POMDPs where there is a collection of MDPs, and each episode, an MDP is sampled from this collection. Although these works do consider system parameters varying between episodes, the primary focus of the experiments usually tends to be on the multi-task setting (i.e. different reward functions instead of transition functions) [77, 18, 60]. We consider not only differing system parameters but also the presence of unmeasured state variables; therefore, the class of POMDPs considered in this paper is broader than the one studied in MetaRL.

Using recurrent networks has long been an approach for tackling POMDPs [29], and is still a common way to do so in a wide variety of settings [19, 73, 70, 50, 47, 75, 66, 11, 2]. Moreover implementations are publicly available both for on-policy [41, 30] and off-policy [50, 75, 11] algorithms, making it an easy pick for those wanting a quick solution. Some works [32, 77, 28, 18, 3] use recurrent networks to estimate the belief state [37], which is a distribution over the agent's true state. However, Ni et al. [50] recently showed that well-implemented, recurrent versions of SAC [26] and TD3 [23] perform competitively with many of these specialized algorithms. In either case, we believe works that estimate the belief state are not in conflict with our own since their architectures can be modified to use GPIDE instead of a recurrent unit.

Beyond recurrent networks, there has been a surge of interest in applying transformers to reinforcement learning [40]. However, we were unable to find many instances of transformers being used as history encoders in the online setting, perhaps because of their difficulty to train. Parisotto et al. [55] introduced a new architecture to remedy these difficulties; however, Melo [46] applied transformers to MetaRL and asserted that careful weight initialization is the only thing needed for stability in training. We note that GPIDE with only attention heads is similar to a single multi-headed self-attention block that appears in many transformer architectures; however, we show that attention is the least important type of head in GPIDE and often hurts performance (see Section 5.3).

Perhaps closest to our proposed architecture is PEARL [60], which does a multiplicative combination of Gaussian distributions corresponding to each state-action-reward tuple. However, their algorithm is designed for the MetaRL setting specifically. Additionally, we note that the idea of summations and averaging has been shown to be powerful in prior works. Specifically, Oliva et al. [52] introduced the Statistical Recurrent Unit, an alternative architecture to LSTMs and GRUs that leverages moving averages and performs competitively across several supervised learning tasks.

There are many facets of RL where improvements can be made to robustness, and many works focus on altering the training procedure. They use techniques such as optimizing the policy's worst-case performance [59, 36] or using variational information bottlenecking (VIB) [4] to limit the information used by the policy [42, 33, 21]. In contrast, our work specifically focuses on how architecture choices of history encoders affect robustness, but we note our developments can be used in conjunctions with

these other directions, possibly resulting in improved robustness. We perform additional experiments that consider VIB in Appendix F.1.

Lastly, we note that there is a plethora of work interested in the intersection of reinforcement learning and PID control [35, 25, 39, 22, 74, 12]. These works focus on using reinforcement learning to tune the coefficients of PID controllers (often in MIMO settings). We view these as important works on how to improve PID control using reinforcement learning; however, we view our own work as how to improve deep reinforcement learning by leveraging ideas from PID control.

## 5 Experiments

In this section, we experimentally compare PIDE and GPIDE against recurrent and transformer encoders. In particular, we explore the following questions:

- How does the performance of a policy using PIDE or GPIDE do on tracking problems? In addition, how well can policies adapt to different system parameters and how robust to modelling error are they on these problems? (Section 5.1)

- Going beyond tracking problems, how well does GPIDE perform on higher dimensional locomotion control tasks (Section 5.2)

- How important is each type of head in GPIDE? (Section 5.3)

For the following tracking problems we use the Soft Actor Critic (SAC) [26] algorithm with each of the different methods for encoding observation history. Following Ni et al. [50], we make two separate instantiations of the encoders for the policy and value networks, respectively. Since the tracking problems are relatively simple, we use a small policy network consisting of 1 hidden layer with 24 units; however, we found that we still needed to use a relatively large Q network consisting of 2 hidden layers with 256 units each to solve the problems. All hyperparameters remain fixed across baselines and tracking tasks; only the history encoders change.

For the recurrent encoder, we use a GRU and follow the implementation of Ni et al. [50] closely. Our transformer encoder closely resembles the GPT2 architecture [58], and it also includes positional encodings for the observation history. For GPIDE, we use $H = 6$ heads: one summation head, two attention heads, and three exponential smoothing heads (with $\alpha = 0.25, 0.5, 1.0$). This choice was not optimized, but rather was picked so that all types of heads were included and so that GPIDE has roughly the same amount of parameters as our GRU baseline. As a reference point for these RL methods, we also evaluate the performance of a tuned PID controller. Not only do PID controllers have an incredibly small number of parameters compared to the other RL-based controllers, but the training procedure is also much more straightforward since it can be posed as a black-box optimization over the returns. While there exists many sophisticated extensions of the PID controller (especially in MIMO systems [8]), we only consider the vanilla PID controller since we believe it serves as a good reference point. All methods are built on top of the rlkit library [57]. More details about implementations, hyperparameters, and computation can be found in Appendices B, C, and D, respectively. We also include additional experiments regarding variational information bottlenecking (VIB) and lookback size ablations in Appendices F.1 and F.2. Implementations can be found at `https://github.com/IanChar/GPIDE`.

### 5.1 Tracking Problems

In this subsection we consider a number of tracking problems. For each environment, the observation consists of the current signals, the reference values, and additional information about the last action made. Unless stated otherwise, the reward is as described in Section 2. More information about environments can be found in Appendix E. To make a fair comparison against PID controls, we choose to only encode the history of observations. For evaluation, we use 100 fixed settings of the environment (each setting consists of targets and system parameters). To avoid overfitting to these 100 settings, we used a separate set of 100 settings and averaged over 3 seeds when developing our methods. We evaluate policies throughout training, but report the average over the last 10% of evaluations as the final returns. We allow each policy to collect one million environment transitions, and all scores are averaged over 5 seeds. Lastly, each table shows scores formed by scaling the returns by the best and worst average returns across all methods in a particular variant of the environment, where scores of 0 and 100 correspond to the worst and best returns respectively.

| Environment (Train/Test) | PID Controller | GRU | Transformer | PIDE | GPIDE |
|---|---|---|---|---|---|
| MSD Fixed/Fixed | $0.00 \pm 3.96$ | $83.73 \pm 3.48$ | $85.79 \pm 1.98$ | $\mathbf{100.00 \pm 0.66}$ | $83.72 \pm 2.86$ |
| MSD Small/Small | $0.00 \pm 5.58$ | $\mathbf{100.00 \pm 1.59}$ | $73.27 \pm 4.98$ | $75.51 \pm 1.31$ | $80.21 \pm 8.59$ |
| MSD Fixed/Large | $36.58 \pm 2.86$ | $0.00 \pm 3.42$ | $\mathbf{53.70 \pm 1.71}$ | $34.92 \pm 0.93$ | $29.55 \pm 2.32$ |
| MSD Small/Large | $43.52 \pm 2.82$ | $\mathbf{87.63 \pm 2.28}$ | $81.44 \pm 0.82$ | $53.21 \pm 1.31$ | $68.03 \pm 4.43$ |
| MSD Large/Large | $45.60 \pm 1.71$ | $\mathbf{100.00 \pm 0.61}$ | $92.60 \pm 1.49$ | $69.88 \pm 0.69$ | $93.03 \pm 1.27$ |
| Average | 25.14 | 74.27 | **77.36** | 66.70 | 70.91 |
| DMSD Fixed/Fixed | $24.33 \pm 3.97$ | $0.00 \pm 8.69$ | $22.05 \pm 3.58$ | $\mathbf{100.00 \pm 1.08}$ | $76.23 \pm 6.26$ |
| DMSD Small/Small | $16.17 \pm 3.09$ | $0.00 \pm 7.79$ | $43.74 \pm 3.70$ | $\mathbf{100.00 \pm 0.94}$ | $86.74 \pm 3.94$ |
| DMSD Fixed/Large | $63.59 \pm 2.91$ | $0.00 \pm 2.28$ | $59.84 \pm 1.13$ | $\mathbf{78.77 \pm 1.16}$ | $63.89 \pm 2.16$ |
| DMSD Small/Large | $70.35 \pm 1.44$ | $39.26 \pm 2.37$ | $73.81 \pm 1.60$ | $88.52 \pm 0.83$ | $\mathbf{89.66 \pm 1.33}$ |
| DMSD Large/Large | $78.77 \pm 1.97$ | $52.01 \pm 2.01$ | $84.45 \pm 1.41$ | $86.90 \pm 0.18$ | $\mathbf{100.00 \pm 0.91}$ |
| Average | 50.64 | 18.25 | 56.78 | **90.84** | 83.30 |
| Total Average | 37.89 | 46.26 | 67.07 | **78.77** | 77.11 |

Table 1: **Mass Spring Damper Task Results**. The scores presented are averaged over five seeds and we show the standard error for each score.

**Mass Spring Damper Tracking**  The first tracking task is the control of a classic 1D toy physics system in which there is a mass attached to a wall by a spring and damper. The goal is then to apply a force to the mass in order to move it to a given reference location. There are three system parameters to consider here: the mass, spring constant, and damping factor. We also consider the substantially more difficult problem in which there are two masses sandwiched between two walls, and the masses are connected to the walls and each other by springs and dampers (see Appendix E.1 for a diagram of this). Overall there are eight system parameters (three spring constants, three damping factors, and two masses) and two actuators (a force applied to each mass). We refer to the first problem as Mass-Spring-Damper (MSD) and the second problem as Double-Mass-Spring-Damper (DMSD).

Additionally, we test how adaptive these policies are by changing system parameters in a MetaRL-type fashion (i.e. for each episode we randomly select system parameters and then fix them for the rest of the episode). Similar to Packer et al. [54], we train the policies on three versions of the environment: one with no variation in system parameters, one with a small amount of variation, and one with a large amount of variation. We evaluate all policies on the version of the environment with large system parameter variation to test generalization capabilities.

Table 1 shows the scores achieved for each of the settings. While GRU and transformers seem to do a good job at encoding history for the MSD environment, both are significantly worse on the more complex DMSD task when compared to our proposed encoders. This is true especially for GRU, which performs worse than two independent PID controllers for every configuration. Additionally, while it seems that GRU can generalize to large amounts of variation in system parameters when a small amount is present, it fails horribly when trained on fixed system parameters. On the other hand, transformers are able to generalize surprisingly well when trained on both fixed system parameters and with small variation. We hypothesize the autoregressive nature of GRU may make it particularly susceptible to overfitting. Comparing PIDE and GPIDE, we see that PIDE tends to shine in the straightforward cases where there is little change in system parameters, whereas GPIDE is able to adapt when there is a large variation in parameters since it has additional capacity.

**Navigation Environment**  To emulate the setting where the policy is trained on an imperfect simulator, we consider an environment in which the agent is tasked with moving itself across a surface to a specified 2D target as quickly and efficiently as possible. At every point in time, the agent can apply some force to move itself, but a penalty term proportional to the magnitude of the force is subtracted from the reward. Suppose that we have access to a simulator of the environment that is perfect except for the fact that it does not model friction between the agent and the surface. We refer to this simulator and the real environment as the "No Friction" and "Friction" environment, respectively. In both environments, the mass of the agent is treated as a system parameter that is sampled for each episode; however, the Friction environment has a larger range of masses and also randomly samples the coefficient of friction each episode.

Figure 2 shows the average returns recorded during training for both navigation environments and when the policies trained in No Friction are evaluated in Friction. A table of final scores can be found in Appendix G.3. One can see that GPIDE not only achieves the best returns in the environments it was trained in, but is also robust when going from the frictionless environment to the one with friction. On the other hand, PIDE has less capacity and therefore cannot achieve the same results; however, it

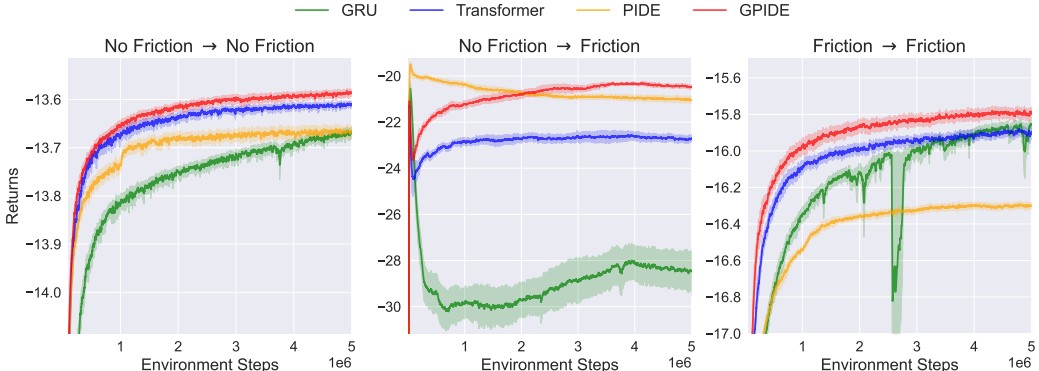

Figure 2: **Average Returns for Navigation Environments.** The curves show the average over five seeds and the shaded region shows the standard error. For this plot, we allowed for 5x the normal amount budget to allow all methods to converge. We omit the PID controllers from this plot since it gets substantially worse returns.

is immediately more robust than the other methods, although it begins to overfit over time. It is also clear that using GRU is less sample efficient and less robust to changes in the test environment.

**Tokamak Control** For our last tracking experiment we return to tokamak control. In particular, we focus on the DIII-D tokamak, a device operated by General Atomics in San Diego, California. We aim to control two quantities: $\beta_N$, the normalized ratio between plasma and magnetic pressure, and rotation, i.e. how fast the plasma is spinning around the toroid. These are important quantities to track because $\beta_N$ serves as an approximate economic indicator and rotation control of the plasma has been suggested to be key for stability [7, 67, 10, 61, 56]. The policy has control over the eight neutral beams [24], which are able to inject power and torque by blasting neutrally charged particles into the plasma. Importantly, two of the eight beams can be oriented in the opposite direction from the others, which decouples the total combined power and torque to some extent (see Figure 3).

To emulate the sim-to-real training experience, we create a simulator based on the equations described in Boyer et al. [9] and Scoville et al. [63]. This simulator has two major shortcomings: it assumes that certain states of the plasma (e.g. its shape) are fixed for entire episodes, and it assumes that there are no events that cause loss of confinement of the plasma. We make up for part of the former by randomly sampling plasma states each episode. The approximate

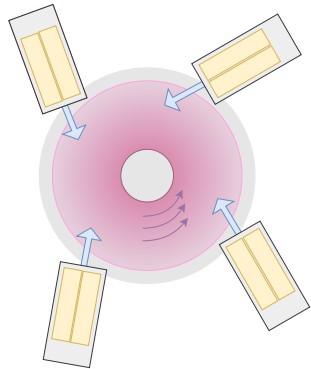

Figure 3: **Illustration of DIII-D from Above.** [14] Each beamline in the figure contains two independent beams (yellow boxes). The plasma is rotating counter-clockwise and the two beams in the bottom left of the figure are oriented in the counter-current direction, allowing power and torque to be decoupled. This figure gives a rough idea of beam positioning but is not physically accurate.

| Environment (Train/Test) | PID Controller | GRU | Transformer | PIDE | GPIDE |
|---|---|---|---|---|---|
| $\beta_N$-Track Sim/Sim | $40.69 \pm 0.32$ | $\mathbf{100.00 \pm 0.20}$ | $97.56 \pm 0.19$ | $0.00 \pm 1.05$ | $98.33 \pm 0.41$ |
| $\beta_N$-Track Sim/Real | $\mathbf{89.15 \pm 0.99}$ | $40.96 \pm 5.45$ | $40.05 \pm 11.91$ | $0.00 \pm 21.04$ | $55.21 \pm 4.44$ |
| $\beta_N$-Track Real/Real | $98.45 \pm 0.77$ | $98.24 \pm 0.38$ | $98.74 \pm 0.29$ | $\mathbf{100.00 \pm 0.23}$ | $99.30 \pm 0.64$ |
| Average | 76.10 | 79.73 | 78.79 | 33.33 | **84.28** |
| $\beta_N$-Rot-Track Sim/Sim | $0.00 \pm 0.83$ | $99.06 \pm 0.22$ | $96.22 \pm 0.94$ | $67.98 \pm 0.50$ | $\mathbf{100.00 \pm 0.29}$ |
| $\beta_N$-Rot-Track Sim/Real | $\mathbf{83.71 \pm 2.64}$ | $39.76 \pm 5.84$ | $33.31 \pm 0.69$ | $0.00 \pm 8.89$ | $51.00 \pm 1.92$ |
| $\beta_N$-Rot-Track Real/Real | $92.02 \pm 0.84$ | $98.34 \pm 0.52$ | $96.32 \pm 0.31$ | $98.21 \pm 0.23$ | $\mathbf{100.00 \pm 0.46}$ |
| Average | 58.58 | 79.05 | 75.28 | 55.40 | **83.67** |
| Total Average | 67.34 | 79.39 | 77.03 | 44.36 | **83.97** |

Table 2: **Tokamak Control Task Results**. The scores presented are averaged over five seeds and we show the standard error for each score.

| Environment | PPO-GRU | TD3-GRU | VRM | SAC-Transformer | SAC-GPIDE |
|---|---|---|---|---|---|
| HalfCheetah-P | $27.09 \pm 7.85$ | $\mathbf{85.80 \pm 5.15}$ | $-107.00 \pm 1.39$ | $37.00 \pm 9.97$ | $82.63 \pm 3.46$ |
| Hopper-P | $49.00 \pm 5.22$ | $84.63 \pm 8.33$ | $3.53 \pm 1.63$ | $59.54 \pm 19.64$ | $\mathbf{93.27 \pm 13.56}$ |
| Walker-P | $1.67 \pm 4.39$ | $29.08 \pm 9.67$ | $-3.89 \pm 1.25$ | $24.89 \pm 14.80$ | $\mathbf{96.61 \pm 1.60}$ |
| Ant-P | $39.48 \pm 3.74$ | $-36.36 \pm 3.35$ | $-36.39 \pm 0.17$ | $-10.57 \pm 2.34$ | $\mathbf{66.66 \pm 2.94}$ |
| HalfCheetah-V | $19.68 \pm 11.71$ | $\mathbf{59.03 \pm 2.88}$ | $-80.49 \pm 2.97$ | $-41.31 \pm 26.15$ | $20.39 \pm 29.60$ |
| Hopper-V | $13.86 \pm 4.80$ | $57.43 \pm 8.63$ | $10.08 \pm 3.51$ | $0.28 \pm 8.49$ | $\mathbf{90.98 \pm 4.28}$ |
| Walker-V | $8.12 \pm 5.43$ | $-4.63 \pm 1.30$ | $-1.80 \pm 0.70$ | $-8.21 \pm 1.31$ | $\mathbf{36.90 \pm 16.59}$ |
| Ant-V | $1.43 \pm 3.26$ | $17.03 \pm 6.55$ | $-13.41 \pm 0.12$ | $0.81 \pm 1.31$ | $\mathbf{18.03 \pm 5.10}$ |
| Average | 20.04 | 36.50 | -28.67 | 7.80 | **63.18** |

Table 3: **PyBullet Task Results.** Each score is averaged over four seeds and we report the standard errors. Unlike before, we scale the returns by the returns of an oracle policy (i.e. one which sees position and velocity) and a policy which does not encode any history. For the environment names, "P" and "V" denote only position or only velocity in the observation, resepctively.

"real" environment addresses these shortcomings by using a data-driven simulator. This approach to simulating has been shown to be relatively accurate [14, 65, 64, 1], and we use an adapted version of the simulator appearing in Char et al. [14] for our work. This simulator accounts for a greater set of the plasma's state, and the additional information is rich enough that loss of confinement events play a role in the dynamics.

We consider two versions of this task: the first is a SISO task where total power is controlled to achieve a $\beta_N$ target, and the second is a MIMO task where total power and torque is controlled to achieve $\beta_N$ and rotation targets. The results for both of these tasks are shown in Table 2. Most of the RL techniques are able to do well if tested in the same environment they were trained in; the exception of this is PIDE, which curiously is unable to perform well in the simulator environment. While no reinforcement learning method matches the robustness of a PID controller, policies trained with GPIDE fare significantly better.

## 5.2 PyBullet Locomotion Tasks

Moving past tracking problems, we evaluate GPIDE on the PyBullet [16] benchmark proposed by Han et al. [28] and adapted in Ni et al. [50]. The benchmark has four robots: halfcheetah, hopper, walker, and ant. For each of these, either the current position information or velocity information is hidden from the agent. Except for GPIDE and transformer encoder, we use all of the performance traces given by Ni et al. [50]. In addition to SAC, they also train using PPO [62], A2C [49], TD3 [23], and VRM [28], a variational method that uses recurrent units to estimate the belief state. We reproduce as much of the training and evaluation procedure as possible, including using the same hyperparameters in the SAC algorithm and giving the history encoders access to actions and rewards. For more information see Appendix C.2. Table 3 shows the performance of GPIDE along with a subset of best performing methods (more results can be found in Appendix G.5). These results make it clear that GPIDE is powerful in arbitrary control tasks besides tracking since the average score achieved across all tasks is a 73% improvement over TD3-GRU, which we believe is the previous state-of-the-art for this benchmark at the time of this work.

Moreover, GPIDE dominates performance for every robot except HalfCheetah. The only setting where GPIDE achieves significantly worse performance is HalfCheetah-V. This setting seemed to cause stability issues in some seeds, lowering the average score. We believe that these stability issues stem from the attention heads, and we found that removing these heads fixed stability issues and resulted in a competitive average score (see Appendix G.5).

## 5.3 GPIDE Ablations

To investigate the role of each type of head, we reran all experiments with three variants of GPIDE: one with six exponential smoothing heads (ES), one with five exponential smoothing heads and

| | MSD | DMSD | Navigation | $\beta_N$ Track | $\beta_N$-Rot Track | PyBullet |
|---|---|---|---|---|---|---|
| ES | +2.69% | -11.14% | -0.11% | +2.57% | +0.29% | +5.81% |
| ES + Sum | -8.33% | +5.49% | -1.65% | +4.22% | +0.76% | +11.00% |
| Attention | -0.36% | -54.95% | -3.91% | -8.85% | -7.55% | -39.44% |

Table 4: **GPIDE Ablation Percent Difference for Average Scores.** All final scores can be found in Appendix G.

one summation head (ES + Sum), and one with six attention heads (see Appendix C.3 for details). We choose these three configurations specifically to better understand the roles that attention and summation play.

Table 4 shows the differences in the average scores for each environment. The first notable takeaway is that having summation is often important in some of the more complex environments. The other takeaway is that much of the heavy lifting is being done by the exponential smoothing. GPIDE fares far worse when only having attention heads, especially in DMSD and the PyBullet environments.

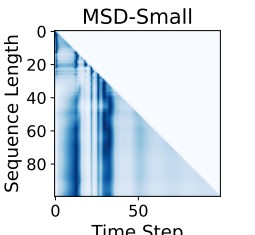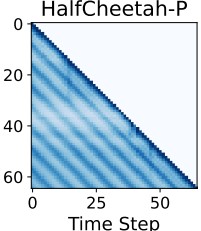

We visualize some of the attention schemes learned by GPIDE for MSD with small variation and HalfCheetah (Figure 4). While the attention scheme learned for MSD could potentially be useful since it recalls information from near the beginning of the episode when the most movement is happening, it appears that the attention scheme for HalfCheetah is simply a poor reproduction of exponential smoothing, making

Figure 4: **Averaged Attention Schemes for MSD-Small and HalfCheetah-P.** Each y-position on the grid corresponds to an amount of history being recorded, and each x-position corresponds to a time point in that history. As such, each of the left-most points are the oldest observation in the history, and the diagonals correspond to the most recent observation. The darker the blue, the greater the weight that is assigned to that time point.

it redundant and suboptimal. In fact, we found this phenomenon to be true across all attention heads and PyBullet tasks. We believe that the periodicity that appears here is due to the oscillatory nature of the problem and lack of positional encoding (although we found including positional encoding degrades performance).

## 6 Discussion

In this work, we introduced the PIDE and GPIDE history encoders to be used for reinforcement learning in partially observable control tasks. Although both are far simpler than prior methods of encoding, they often result in powerful yet robust controllers. We hope that this work inspires the research community to think about how pre-existing control methods can inform architecture choices.

**Limitations** There are many different ways a control task may be partially observable, and we do not believe that our proposed methods are solutions to all of them. For example, we do not think GPIDE is necessarily suited for tasks where the agent needs to remember events (e.g. picking up a key to unlock a door).

As with any bias, the PID-inspired biases that we propose in this work come at the cost of flexibility. For the experiments considered in this work, this trade off is beneficial and results in better policies. However, it is unclear whether this trade off is always worth making. It is possible that in higher dimensional environments or environments with more complex dynamics that having more flexibility is preferable to our proposed architecture.

Lastly, some tasks may require the policy to act on images as observations. We are optimistic that PIDE and GPIDE are still useful architectures in this setting, but we speculate that this is contingent on training an image encoder that is well-suited for these architectures, and we leave this research direction for future work.

## 7 Acknowledgements

We would like to thank Conor Igoe for his helpful discussions and advice on this work. We would also like to thank the NeurIPS 2023 reviewers assigned to our paper for their discussion and feedback.

This work was supported in part by US Department of Energy grants under contract numbers DE-SC0021414 and DE-AC02-09CH1146. This work is also supported by the National Science Foundation Graduate Research Fellowship Program under Grant No. DGE1745016 and DGE2140739. Any opinions, findings, and conclusions orrecommendations expressed in this material are those of the author(s) and do not necessarily reflect the views of the National Science Foundation.

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

# Appendices

## Contents

# A    Additional GPIDE Details

## A.1    Forming the P, I, and D Features with GPIDE

As mentioned in Section 3, the P, I, and D features associated with a PID controller can be easily formed with GPIDE, and in this Appendix we show this concretely. Assume that the observations take the form $o_t = \left( x_t^{(1)}, \ldots, x_t^{(M)}, \sigma_t^{(1)}, \ldots, \sigma_t^{(M)} \right)$, and assume that the actions and rewards are not given to GPIDE for simplicity. Then the linear projection for each head, $f_\theta^h$, takes in $x_{t-1}^{(1)}, \ldots, x_{t-1}^{(M)}, \sigma_{t-1}^{(1)}, \ldots, \sigma_{t-1}^{(M)}, (x_t^{(1)} - x_{t-1}^{(1)}), \ldots, (x_t^{(M)} - x_{t-1}^{(M)}), (\sigma_t^{(1)} - \sigma_{t-1}^{(1)}), \ldots, (\sigma_t^{(M)} - \sigma_{t-1}^{(M)})$ at each time step $t$. We use a GPIDE architecture with three heads and where $g_\theta$ is the identity function. The linear projections for each head is as follows:

$$
f_\theta^1(o_{t-1}, o_t - o_{t-1}) = f_\theta^2(o_{t-1}, o_t - o_{t-1}) = \begin{bmatrix} (x_t^{(1)} - x_{t-1}^{(1)}) + x_{t-1}^{(1)} - (\sigma_t^{(1)} - \sigma_{t-1}^{(1)}) + \sigma_{t-1}^{(1)} \\ \vdots \\ (x_t^{(M)} - x_{t-1}^{(M)}) + x_{t-1}^{(M)} - (\sigma_t^{(M)} - \sigma_{t-1}^{(M)}) + \sigma_{t-1}^{(M)} \end{bmatrix}
$$

$$
f_\theta^3(o_{t-1}, o_t - o_{t-1}) = \begin{bmatrix} (x_t^{(1)} - x_{t-1}^{(1)}) - (\sigma_t^{(1)} - \sigma_{t-1}^{(1)}) \\ \vdots \\ (x_t^{(M)} - x_{t-1}^{(M)}) - (\sigma_t^{(M)} - \sigma_{t-1}^{(M)}) \end{bmatrix}
$$

Note that $f_\theta^1$ and $f_\theta^2$ form the current error at time $t$ (i.e. the P term), and $f_\theta^3$ forms the change in error (i.e. the D term). For accumulation strategies, using exponential smoothing with $\alpha = 1$ for the first and third heads and a summation head for the second head will recover the P, I, and D terms for heads 1, 2, and 3, respectively. Note that the above assumes $dt = 1$, but the linear projections can be adjusted to take different $dt$ values into account.

# B    Implementation Details

**Code Release** All code for implementations are provided in the supplemental material along with instructions for how to run experiments. The code can also be found at `https://github.com/IanChar/GPIDE`. The only experiment that cannot be run are the "real" cases for tokamak control.

**Architecture**   We use the same general architecture for each of the RL methods in this paper (see Figure 5). Each input to the history encoders, policy functions, and $Q$-value functions have corresponding encoders. This setup closely follows what was done in Ni et al. [50]. The encoders are simply linear projections; however, in the case of our GRU history encoder we do linear projections followed by a ReLU activation (as done in Ni et al. [50]). Although hypothetically the policy only needs to take in history encoding, $z_t$, since int includes the current observation, we found it essential for the current observation to be passed in independently and have its own encoder.

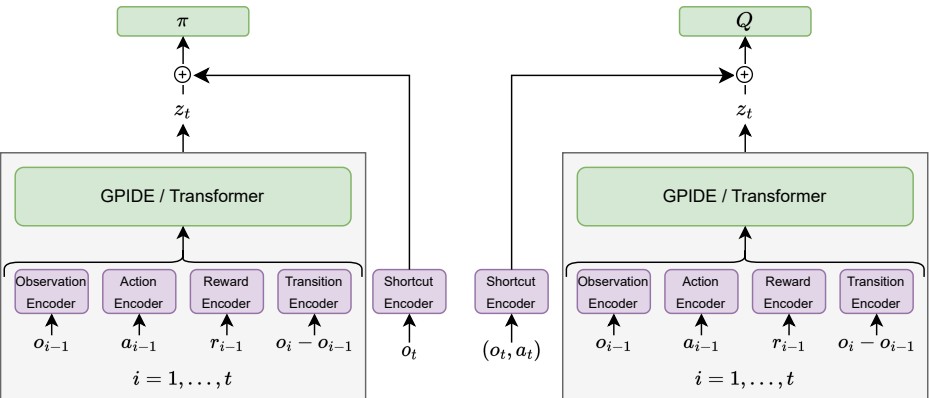

Figure 5: **General Policy and Q Function Architectures.** This architecture is heavily inspired by Ni et al. [50]. The gray box shows the history encoder modules, and this is the only thing that changes between baseline methods in the tracking problems. Note that there are two encoders: one for the policy function and one for the $Q$ value function. The purple boxes show the input encoders, and hyperaparmeters for these can be found in Table 6. We found the shortcut encoders to be essential to good performance. The architecture when using GRU is nearly identical; however, there is no "Transition Encoder" since Ni et al. [50] encodes $(o_i, a_{i-1}, r_{i-1})$ for each time step instead.

## B.1 GPIDE Implementation Details

In addition to what is mentioned in Section 3, we found that there were several choices that helped with training. First, there may be some scaling issues because $o_t - o_{t-1}$ may be small or the result of summation type heads may result in large encodings. To account for this, we use batch normalization layers [34] before each input encoding and after each $\ell^h$.

There are very few nonlinear components of GPIDE. The only one that remains constant across all experiments is that a tanh activation is used for the final output of the encoder. For tracking tasks, the decoder $g_\theta$ has 1 hidden layer with 64 units and uses a ReLU activation function. For PyBullet tasks, $g_\theta$ is a linear function.

## B.2 Recurrent and Transformer Baseline Details

**Recurrent Encoder.** For the recurrent encoder, we tried to match as many details as Ni et al. [50] as possible. We double checked our implementation against theirs and confirmed that it achieves similar performance.

**Transformer Encoder.** We follow the GPT2 architecture [58] for inspiration, and particularly the code provided in Karpathy [38]. In particular, we use a number of multi-headed self-attention blocks in sequence with residual connections. We use layer normalization [6] before multi-headed attention and out projections; however, we do not use dropout. The out projection for each multi-headed self-attention block has one hidden layer with four times the number of units as the embedding dimension. Although Melo [46] suggests using T-Fixup weight initialization, we found that more reliably high performance was achieved with the weight initialization of Radford et al. [58]. Lastly, we used the same representation for the history as GPIDE, i.e. $(o_{t-1}, a_{t-1}, r_{t-1}, o_t - o_{t-1})$, since it results in better performance.

## B.3 PID Baseline

To tune our PID baseline, we used Bayesian Optimization over the three (for SISO) or six (for MIMO) dimensional space. Specifically we use the library provided by Nogueira [51]. The output of the blackbox optimization is the average over 100 different settings (independent from the 100 settings used for testing). We allow the optimization procedure to collect as many samples as the RL methods. The final performance reported uses the PID controller with the best gains found during the

optimization procedure. The bounds for each of the tracking tasks were eyeballed to be appropriate, which potentially preferably skews performance.

## C   Hyperparameters

Because of resource restrictions, we were unable to do full hyperparameter tuning for each benchmark presented in this paper. Instead, we focused on ensuring that all history encoding methods were roughly comparable, e.g. dimension of encoding, number of parameters, etc. Tables 5 show selected hyperparameters, and the following subsections describe how an important subset of these hyperparameters were picked. Any tuning that was done was over three seeds using 100 fixed settings (different from the 100 settings used for testing).

| Task Type | Learning Rate | Batch Size | Discount Factor | Policy Network | Q Network | Path Length Encoding |
|---|---|---|---|---|---|---|
| Tracking | $3e^{-4}$ | 32 (256 for PIDE) | 0.95 | [24] | [256, 256] | 100 |
| PyBullet | $3e^{-4}$ | 32 (256 for PIDE) | 0.99 | [256, 256] | [256, 256] | 64 |

Table 5: **SAC Hyperparameters.** The "Path Length Encoding" is the amount of history each encoder gets to observe besides PIDE which, because of the nature of it, uses the entire episode.

| | Observation | Action | Reward | Transition | Policy Shortcut | $Q$ Shortcut | History Encoding |
|---|---|---|---|---|---|---|---|
| GPIDE (Tracking) | 8 | N/A | N/A | 8 | 8 | 64 | 64 |
| GRU (Tracking) | 8 | N/A | N/A | N/A | 8 | 64 | 64 |
| Transformer (Tracking) | 16 | N/A | N/A | 16 | 8 | 64 | 64 |
| GPIDE (PyBullet) | 32 | 16 | 16 | 64 | 8 | 64 | 128 |
| Transformer (PyBullet) | 48 | 16 | 16 | 48 | 8 | 64 | 128 |

Table 6: **Dimension for the Input Encoders and Final History Encoding**. The input encoders correspond to the output dimensions of the purple boxes in Figure 5. By "History Encoding" size we mean the dimension of $z_t$.

| Task Type | $D$ | $g_\theta$ Hidden Size |
|---|---|---|
| Tracking | 16 | [64] |
| PyBullet | 32 | [] |

Table 7: **GPIDE Specific Hyperparamters.** Recall that $D$ corresponds to the output dimension of $f_\theta$. Empty brackets for the hidden size means that $g_\theta$ is a linear function.

### C.1   Hyperparamters for Tracking Tasks

For tracking tasks, we tried using a history encoding size of 32 and 64 for GRU, and we found that performance was better with 64. This is surprising since PIDE can perform well in these environments even though its history encoding is much smaller (3 or 6 dimensional). To make it a fair comparison, we set the history encoding dimension for GPIDE and transformer to be 64 as well. We use one layer for GRU. For the transformer-specific hyperparameters we choose half of what appears in the PyBullet tasks.

### C.2   Hyperparameters for PyBullet Task

For the PyBullet tasks, we simply tried to emulate most of the hyperparameters found in Ni et al. [50]. For the transformer, we choose to use similar hyperparameters to those found in Melo [46]. However, we found that, unlike the tracking tasks, positional encoding hurts performance. As such, we do not include it for PyBullet experiments.

### C.3   Hyperparameters for Ablations

For the ablations of GPIDE, we use $\alpha = 0.01, 0.1, 0.25, 0.5, 0.9, 1.0$ for the smoothing parameters when only exponential smoothing is used. When using exponential smoothing and summation, the $\alpha = 0.01$ head is replaced with a summation head. The attention version of GPIDE replaces all six of these heads with attention.

| Task Type | Number of Layers | Number of Heads | Embedding Size per Head |
|-----------|:----------------:|:---------------:|:-----------------------:|
| Tracking  | 2 | 4 | 8 |
| PyBullet  | 4 | 8 | 16 |

Table 8: **Transformer Specific Hyperparamters**

| Encoder | SISO Tracking | MIMO Tracking (2D) | PyBullet |
|---------|:-------------:|:------------------:|:--------:|
| Transformer | 25,542 | 25,644 | 793,868-795,026 |
| GRU | 14,240 | 14,264 | 74,816-75,248 |
| GPIDE | 13,228 | 13,288 | 75,296-76,486 |
| GPIDE-ES | 12,204 | 12,264 | 50,720-51,910 |
| GPIDE-ESS | 12,204 | 12,264 | 50,720-51,910 |
| GPIDE-Attention | 15,276 | 15,336 | 99,872-101,062 |

Table 9: **Number of Parameters in History Encoder Modules**. The number of parameters corresponds to the gray boxes in Figure 5. The difference in SISO vs MIMO and the PyBullet tasks is due to the different observation and action space dimensionalities.

## D   Computation Details

We used an internal cluster of machines to run these experiments. We mostly leveraged Nvidia Titan X GPUs for this, but also used a few Nvidia GTX 1080s. It is difficult to get an accurate estimate of run time since job loads vary drastically on our cluster from other users. However, to train a single policy on DMSD to completion (1 million transitions collected, or 1,000 epochs) using PIDE takes roughly 4.5 hours, using GPIDE takes roughly 17.25 hours, using a GRU takes roughly 14.5 hours, and using a transformer takes roughly 21 hours. This is similar for other tracking tasks. For PyBullet tasks, using GPIDE took roughly 43.2 hours and using a transformer took roughly 64.2 hours. We note that our implementation of GPIDE is somewhat naive and could be vastly improved. In particular, for exponential smoothing and summation heads, $w_t$ can be cached to save on compute, which is not being done currently. This is a big advantage in efficiency that GPIDE (especially one without attention heads) has over transformers.

## E   Environment Descriptions

### E.1   Mass Spring Damper

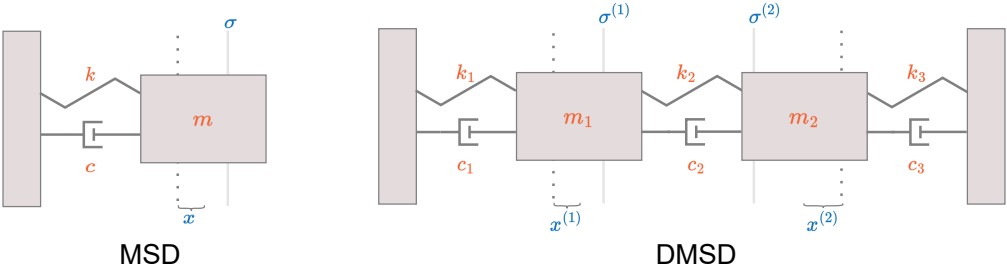

Figure 6: **Diagram of the Mass Spring Damper Environments.** The diagram on the left the Mass Spring Damper (MSD) environment, and the diagram on the right shows the Double Mass Spring Damper (DMSD) environment. In the diagram, we have labelled the system parameters and the parts of the observation. The dotted line shows where the center of the mass is located with no force applied, and the current position of the mass is measured with respect to this point.

For both MSD and DMSD, the observations include the current mass position(s), the target reference position(s), and the last action played. Each episode lasts for 100 time steps. For all RL methods, the action is a difference in force applied to the mass, but for the PID the action is simply the force

| System Parameter | Fixed | Small | Large |
|---|---|---|---|
| Damping Constant | $\mathcal{U}(4.0, 4.0)$ | $\mathcal{U}(3.5, 5.5)$ | $\mathcal{U}(2.0, 10.0)$ |
| Spring Constant | $\mathcal{U}(2.0, 2.0)$ | $\mathcal{U}(1.75, 3.0)$ | $\mathcal{U}(0.5, 6.0)$ |
| Mass | $\mathcal{U}(20.0, 20.0)$ | $\mathcal{U}(17.5, 40.0)$ | $\mathcal{U}(10.0, 100.0)$ |

Table 10: **MSD and DMSD System Parameter Distributions.** Each episode system parameters are uniformly at random drawn from these bounds.

| System Parameter | No Friction | Friction |
|---|---|---|
| Total Friction | $\mathcal{U}(0.0, 0.0)$ | $\mathcal{U}(0.05, 0.25)$ |
| Static Friction (Proportion) | $\mathcal{U}(0.0, 0.0)$ | $\mathcal{U}(0.25, 0.75)$ |
| Mass | $\mathcal{U}(15.0, 25.0)$ | $\mathcal{U}(5.0, 35.0)$ |

Table 11: **Navigation System Parameter Distributions.** Each episode system parameters are uniformly at random drawn from these bounds. The static friction parameter drawn is the proportion of the total friction that is static friction.

to be applied to the mass at that time. The force is bounded between -10 and 10 $N$ for MSD and -30 and 30 $N$ for DMSD. Each episode, system parameters are drawn from a uniform distribution with bounds shown in Table 10 (they are the same for both MSD and DMSD). Targets are drawn to uniformly at random to be $-1.5$ to $1.5$ $m$ offset from the masses' resting positions.

### E.2 Navigation Environment

Like the MSD and DMSD environments, the navigation experiment lasts 100 time steps each episode. Additionally, the observation includes position signal, target locations, and the last action. For all methods we set the action to be the change in force, and the total amount of force is bounded between -10 and 10 $N$. The penalty on the reward is equal to 0.01 times the magnitude of the change in force. In addition, the maximum magnitude of the velocity for the agent is bounded by $1.0 m/s$. The agent always starts at the location $(0, 0)$, and the target is picked uniformly at random to be within a box of length 10 centered around the origin.

Every episode, the mass, kinetic friction coefficient, and static friction coefficient is sampled, The friction is sampled by first sampling the total amount of friction in the system, and then sampling what proportion of the total friction is static friction. All distributions for the system parameters are uniform, and we show the bounds in Table 11.

### E.3 Tokamak Control Environment

**Simulator**   Our simulator version of the tokamak control is inspired by equations used by Boyer et al. [9], Scoville et al. [63]. In particular, we use the following relations for stored energy, $E$, and rotation, $v_{\text{rot}}$:

$$\dot{E} = P - \frac{E}{\tau_E}$$

$$\dot{v}_{\text{rot}} = C_{\text{rot}}T - \frac{v_{\text{rot}}}{\tau_m}$$

where $P$ is the total power, $T$ is the total torque, $\tau_E$ is the energy confinement time, $\tau_m$ is the momentum confinement time, and $C_{\text{rot}}$ is a quantity relying on the ion density and major radius of the plasma. We treat $\tau_m$ and $C_{\text{rot}}$ as constants with values of 0.1 and 80.0 respectively.

We base the energy confinement time off of the ITERH-98 scaling [68]. This uses many measurements of the plasma, but we focus on a subset of these and treat the rest as constants. In particular,

$$\tau_E = C_E I^{0.95} B^{0.15} P^{-0.69}$$

where $C_E$ is a constant value we set to be 200, $I$ is the plasma current, and $B$ is the toroidal magnetic field. To relate the stored energy to $\beta_N$ we use the rough approximation

$$\beta_N = C_\beta \left(\frac{aB}{I}\right) E$$

| Minor Radius (m) | Plasma Current (MA) | Toroidal Magnetic Field (T) |
|---|---|---|
| $\mathcal{N}(0.589, 0.02)$ | $\mathcal{N}(1e6, 1e5)$ | $\mathcal{N}(2.75, 0.1)$ |

Table 12: **Tokamak Control Simulator Distributions.**

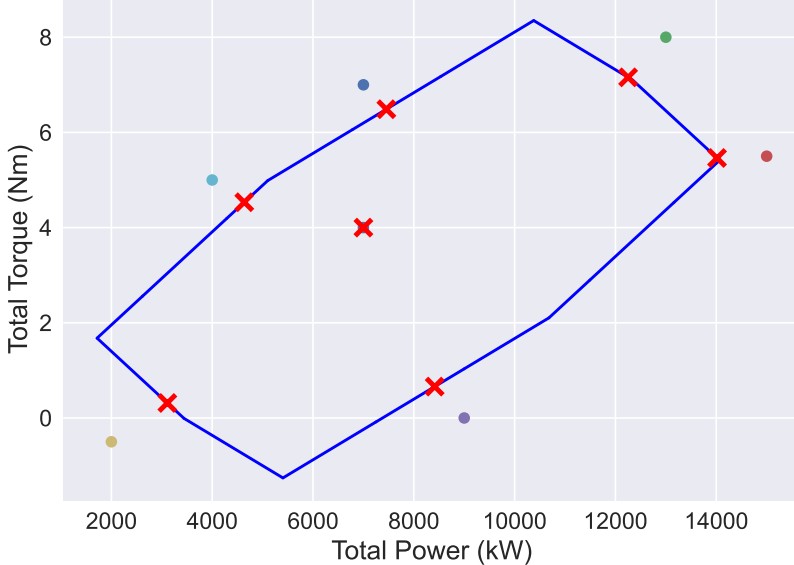

Figure 7: **Power and Torque Bounds.** The region outlined in blue shows the possible power-torque configurations. The dots show possible requests, and the corresponding red $X$ marks show the actual achieved power-torque setting.

where $C_\beta$ is a constant we set to be 5, and $a$ is the minor radius of the plasma. For $a$, $I$, and $B$, we sample these from the distribution described in Table 12 for each episode. Lastly, we add momentum to the stored energy. That is, the stored energy derivative at time $t$, $\dot{E}_t$, is

$$\dot{E}_t = 0.5 \left( P_t - \frac{E_t}{\tau_E} \right) + 0.5 \dot{E}_{t-1}$$

The actions for all control methods is the amount of change for the power and torque. Because the total amount of power and torque injected rely on the beams, they are not totally disentangled. In Figure 7, we show the bounds for the action space. Furthermore, we bound the amount that power and torque can be changed by roughly $40MW/s$ and $35Nm/s$, respectively. Each step is $0.025$ seconds.

Each episode lasts for 100 increments of $0.025$ seconds. The observations are the current $\beta_N$ and rotation values, their reference values, and the current power and torque settings. We make the initial $\beta_N$ and rotation relatively small in order to simulate the plasma ramping up. We let the $\beta_N$ and rotation targets be distributed as $\mathcal{U}(1.75, 2.75)$ and $\mathcal{U}(25.0, 50.0)$ $rad/s$, respectively.

**"Real"** For the real versions of the tokamak control experiments, most of the previous (such as action bounds and target distributions) stays the same. This data-driven simulator is based on the one from Char et al. [14], and we refer the reader there for more details. That being said, there are some differences between the architecture presented there and the one used in this work. Our network is a recurrent network that uses a GRU, has four hidden layers with 512 units each, and outputs the mean and log variance of a normal distribution describing how $\beta_N$ and rotation will change. In addition to power and torque, it takes in measurements for the plasma current, the toroidal magnetic field, n1rms (a measurement related to the plasma' stability), and 13 other actuator requests for gas control and plasma shaping. In addition to sampling from the normal distribution outputted by the network, we train an ensemble of ten networks, and an ensemble member is selected every episode. We use five of

these models during training and the other five during testing. Along with an ensemble member being sampled each episode, we also sample a historical run, which determines the starting conditions of the plasma and how the other inputs to the neural network which are not modelled evolve over time. Recall that 100 fixed settings are used to evaluate the policy every epoch of training. In this case, a setting consists of targets, an ensemble member, and a historical run.

# F   Additional Experiments

## F.1   Experiments Using VIB + GRU

As shown in this work, using a GRU for a history encoder often results in a policy that is ill-equipped to handle changes in the dynamics not seen at train time. One may wonder whether using other robust RL techniques is able to mask this inadequacy of GRU. To test this, we look at adding Variational Information Bottlenecking (VIB) to our GRU baseline [4]. Previous works applying this concept to RL usually do not consider the same class of POMDPs as us [42, 33]; however, Eysenbach et al. [21] does have a baseline that uses VIB with a recurrent policy.

To use VIB with RL, we alter the policy network so that it encodes input to a latent random variable, and the decodes into an action. Following the notation of Lu et al. [42], let this latent random variable be $Z$ and the random variable representing the input of the network be $S$. The goal is to learn a policy that maximizes $J(\pi)$ subject to $I(Z, S) \leq I_C$, where $I(Z, S)$ is the mutual information between $Z$ and $S$, and $I_C$ is some given threshold. In practice, we optimize the Lagrangian. Where $\beta$ is a Lagrangian multiplier, $p(Z|S)$ is the conditional density of $Z$ outputted by the encoder, and $q(Z)$ is the prior, the penalizer is $-\beta \mathbb{E}_S \left[ D_{\text{KL}} \left( p(Z|S) || q(Z) \right) \right]$. Like other works, we assume that $q(Z)$ is a standard multivarite normal.

We alter our GRU baseline for tracking tasks so that the policy uses VIB. This is not entirely straightforward since our policy network is already quite small. We choose to keep as close to original policy architecture as possible and set the dimension of the latent variable, $Z$, to be 24. Note that this change has no affect on the history encoder; this only affects the policy network. For our experiments, we set $\beta = 0.1$, but we note that we may be able to achieve better performance through more careful tuning or annealing of $\beta$.

In any case, we do see that VIB helps with robustness in many instances (see Table 13). However, the cases where there are improvements are instances where the GRU policy already did a good job at generalizing to the test environment. These are primarily the MSD and DMSD environments where the system parameters drawn during training time are simply a subset of those drawn during testing time (interestingly, this notion of dynamics generalization matches the set up of the experiments presented in Lu et al. [42]). Surprisingly, in the navigation and tokamak control experiments, where there are more complex differences between the train and test environments, VIB can sometimes hurt the final performance.

| | PID Controller | GRU | GRU+VIB | Transformer | PIDE | GPIDE |
|---|---|---|---|---|---|---|
| MSD Fixed / Fixed | $-6.14 \pm 0.02$ | $-5.76 \pm 0.02$ | $-5.73 \pm 0.01$ | $-5.75 \pm 0.01$ | $\mathbf{-5.69 \pm 0.00}$ | $-5.76 \pm 0.01$ |
| MSD Fixed / Large | $-11.39 \pm 0.09$ | $-12.52 \pm 0.11$ | $-12.50 \pm 0.14$ | $\mathbf{-10.87 \pm 0.05}$ | $-11.44 \pm 0.03$ | $-11.61 \pm 0.07$ |
| MSD Small / Small | $-7.49 \pm 0.03$ | $-7.02 \pm 0.01$ | $\mathbf{-7.01 \pm 0.01}$ | $-7.15 \pm 0.02$ | $-7.14 \pm 0.01$ | $-7.12 \pm 0.04$ |
| MSD Small / Large | $-11.18 \pm 0.09$ | $-9.82 \pm 0.07$ | $\textcolor{green}{-9.57 \pm 0.03}$ | $-10.01 \pm 0.03$ | $-10.88 \pm 0.04$ | $-10.43 \pm 0.14$ |
| DMSD Fixed / Fixed | $-15.33 \pm 0.14$ | $-16.20 \pm 0.31$ | $\textcolor{green}{-15.83 \pm 0.28}$ | $-15.41 \pm 0.13$ | $\mathbf{-12.64 \pm 0.04}$ | $-13.49 \pm 0.22$ |
| DMSD Fixed / Large | $-27.59 \pm 0.44$ | $-37.21 \pm 0.35$ | $\textcolor{green}{-35.34 \pm 0.28}$ | $-28.16 \pm 0.17$ | $\mathbf{-25.29 \pm 0.18}$ | $-27.54 \pm 0.33$ |
| DMSD Small / Small | $-21.78 \pm 0.14$ | $-22.49 \pm 0.34$ | $-22.51 \pm 0.24$ | $-20.56 \pm 0.16$ | $\mathbf{-18.09 \pm 0.04}$ | $-18.67 \pm 0.17$ |
| DMSD Small / Large | $-26.57 \pm 0.22$ | $-31.27 \pm 0.36$ | $-30.93 \pm 0.34$ | $-26.04 \pm 0.24$ | $-23.82 \pm 0.13$ | $\mathbf{-23.65 \pm 0.20}$ |
| Nav Sim / Sim | $-17.23 \pm 0.18$ | $-13.82 \pm 0.01$ | $-14.69 \pm 0.02$ | $-13.68 \pm 0.01$ | $-13.74 \pm 0.00$ | $\mathbf{-13.65 \pm 0.00}$ |
| Nav Sim / Real | $-23.87 \pm 0.29$ | $-29.85 \pm 0.55$ | $\textcolor{red}{-39.57 \pm 0.24}$ | $-22.84 \pm 0.11$ | $\mathbf{-20.37 \pm 0.08}$ | $-21.23 \pm 0.12$ |
| $\beta_N$ Sim / Sim | $-8.09 \pm 0.00$ | $\mathbf{-7.19 \pm 0.00}$ | $-7.24 \pm 0.01$ | $-7.22 \pm 0.00$ | $-8.71 \pm 0.02$ | $-7.21 \pm 0.01$ |
| $\beta_N$ Sim / Real | $\mathbf{-16.41 \pm 0.30}$ | $-31.21 \pm 1.67$ | $-32.19 \pm 1.19$ | $-31.49 \pm 3.66$ | $-43.78 \pm 6.46$ | $-26.83 \pm 1.36$ |
| $\beta_N$-Rotation Sim / Sim | $-27.56 \pm 0.08$ | $-18.53 \pm 0.02$ | $-18.61 \pm 0.12$ | $-18.79 \pm 0.09$ | $-21.36 \pm 0.05$ | $\mathbf{-18.45 \pm 0.03}$ |
| $\beta_N$-Rotation Sim / Real | $\mathbf{-30.08 \pm 0.95}$ | $-45.91 \pm 2.10$ | $-44.24 \pm 1.33$ | $-48.23 \pm 0.25$ | $-60.23 \pm 3.20$ | $-41.86 \pm 0.69$ |
| Average | -18.33 | -20.12 | -21.14 | -18.71 | -19.58 | -17.51 |

Table 13:   **Tracking Experiments with GRU+VIB**. We use green and red text to highlight significant improvements and deteriorations in performance over vanilla GRU. We only highlight a subset of configurations since we are focused on the robustness properties. This table shows average (unnormalized) returns.

## F.2 Lookback Size Ablations

To better understand the role of the maximum lookback size (i.e. the amount of history used to form the encoding) of GPIDE, we repeat the PyBullet experiments using a lookback size of 4, 16, 64, and 128 with and without attention (labelled GPIDE and GPIDE-ESS, respectively). Figures 8 and 9 show performance curves for GPIDE and GPIDE-ESS respectively. It is clear that there is a massive increase in improvement when expanding the maximum size of the lookback from 4 to 16. For the most part, this trend continues expanding the lookback size from 16 to 64; however, it seems pushing from 64 to 128 yields mixed results. For some tasks, such as HalfCheetah-P, expanding the lookback to 128 results in noticeable improvements both with and without attention. For other tasks, such as Hopper-V, this expansion yields slightly worse performance, possibly because of training stability issues.

One interesting observation is that, when attention is included, this decrease in performance from expanding lookback can occur when increasing from 16 to 64 (see HalfCheetah-V and Walker-V). At the same time, however, it appears the GPIDE can sometimes maintain good performance when expanding the lookback to 128 when GPIDE-ESS cannot (Walker-P).

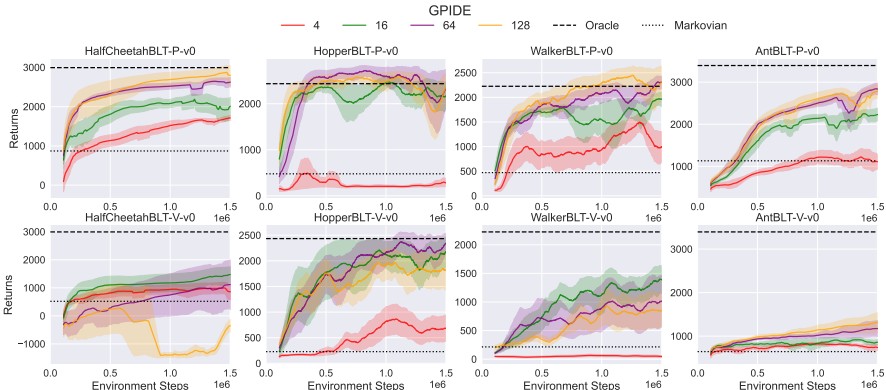

Figure 8: **GPIDE Lookback Ablation.** Each curve shows the average over four seeds and the standard error of each.

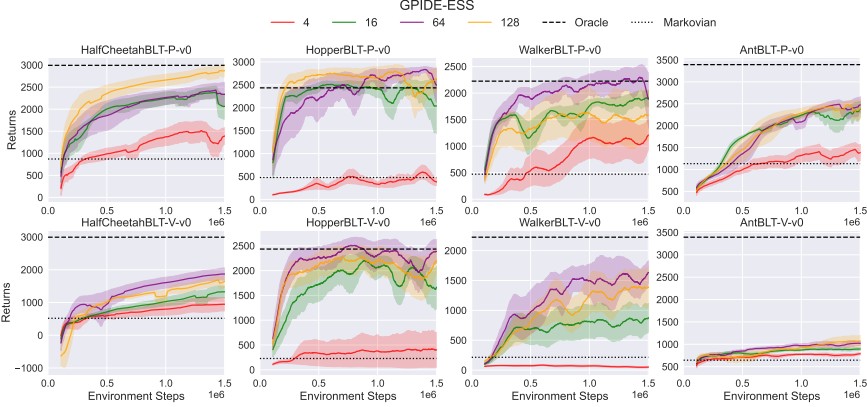

Figure 9: **GPIDE-ESS Lookback Ablation.** Each curve shows the average over four seeds and the standard error of each.

# G   Further Results

In this Appendix, we give further evaluation of the evaluation procedure. In addition, we give full tables of results for normalized and unnormalized scores for all methods. We also show performance

traces. Note that the percentage changes in Table 4 do not necessarily reflect tables in this section since they report all combinations of environment variants.

## G.1 Evaluation Procedure

As stated in the main paper, for tracking tasks, we fix 100 settings (each comprised of targets, start state, and system parameters) that are used to evaluate the policy for every epoch of training (i.e. for every epoch the evaluation returns is the average over all 100 settings returns). We use a separate 100 settings when tuning. For the final returns, we average over the last 10% of recorded evaluations.

For the PyBullet tasks, we use ten different rollouts for evaluation following Ni et al. [50]. We also average over the last 20% of recorded evaluations like they do.

**Normalized Table Scores.** We now give an in-depth explanation of how the scores in the table are computed. Let $\pi_{(b,i)}$ be the policy trained with baseline method $b$ (e.g. with GPIDE, transformer, or GRU encoder) on environment variant $i$ (e.g. fixed, small, or large). Let $J_j(\pi_{(b,i)})$ be the evaluation of policy $\pi_{(b,i)}$ on environment variant $j$, i.e. the average returns over all seeds and episodes. The normalized score for policy $\pi_{(b,i)}$ on variant $j$ is then

$$\frac{J_j(\pi_{(b,i)}) - \min_{b',i'} J_j(\pi_{(b',i')})}{\max_{b',i'} J_j(\pi_{(b',i')}) - \min_{b',i'} J_j(\pi_{(b',i')})}$$

Note that we only min and max over baseline methods presented in the table.

For PyBullet tasks, we do the same procedure but normalize by the oracle policy's performance (sees both position and velocity and has no history encoder) and the Markovian policy's performance (sees only position or velocity and has no history encoder). For both of these policies, we use what was reported from Ni et al. [50]. Note the our normalized scores differ slightly from those used in Ni et al. [50] since they normalize based on the best and worst returns of any policy; however, we believe our scheme gives a more intuitive picture of how any given policy is performing.

## G.2 MSD and DMSD Results

| | PID Controller | GRU | Transformer | PIDE | GPIDE | GPIDE-ES | GPIDE-ESS | GPIDE-Attn |
|---|---|---|---|---|---|---|---|---|
| Fixed / Fixed | $-6.14 \pm 0.02$ | $-5.76 \pm 0.02$ | $-5.75 \pm 0.01$ | $\mathbf{-5.69 \pm 0.00}$ | $-5.76 \pm 0.01$ | $-5.75 \pm 0.01$ | $-5.73 \pm 0.01$ | $-5.83 \pm 0.02$ |
| Fixed / Small | $-7.51 \pm 0.04$ | $-7.56 \pm 0.03$ | $\mathbf{-7.29 \pm 0.01}$ | $-7.37 \pm 0.01$ | $-7.33 \pm 0.04$ | $-7.37 \pm 0.01$ | $-7.32 \pm 0.03$ | $-7.39 \pm 0.03$ |
| Fixed / Large | $-11.39 \pm 0.09$ | $-12.52 \pm 0.11$ | $\mathbf{-10.87 \pm 0.05}$ | $-11.44 \pm 0.03$ | $-11.61 \pm 0.07$ | $-11.48 \pm 0.05$ | $-12.50 \pm 0.19$ | $-11.52 \pm 0.10$ |
| Small / Fixed | $-6.26 \pm 0.06$ | $\mathbf{-5.80 \pm 0.00}$ | $-5.92 \pm 0.01$ | $-5.95 \pm 0.01$ | $-5.93 \pm 0.05$ | $-5.89 \pm 0.01$ | $-5.92 \pm 0.02$ | $-5.91 \pm 0.02$ |
| Small / Small | $-7.49 \pm 0.03$ | $\mathbf{-7.02 \pm 0.01}$ | $-7.15 \pm 0.02$ | $-7.14 \pm 0.01$ | $-7.12 \pm 0.04$ | $-7.09 \pm 0.02$ | $-7.15 \pm 0.02$ | $-7.12 \pm 0.02$ |
| Small / Large | $-11.18 \pm 0.09$ | $\mathbf{-9.82 \pm 0.07}$ | $-10.01 \pm 0.03$ | $-10.88 \pm 0.04$ | $-10.43 \pm 0.14$ | $-10.42 \pm 0.13$ | $-10.43 \pm 0.12$ | $-10.07 \pm 0.14$ |
| Large / Fixed | $-6.78 \pm 0.16$ | $\mathbf{-6.08 \pm 0.01}$ | $-6.28 \pm 0.03$ | $-6.27 \pm 0.01$ | $-6.27 \pm 0.03$ | $-6.23 \pm 0.04$ | $-6.25 \pm 0.04$ | $-6.28 \pm 0.05$ |
| Large / Small | $-7.78 \pm 0.12$ | $\mathbf{-7.25 \pm 0.02}$ | $-7.44 \pm 0.05$ | $-7.43 \pm 0.02$ | $-7.45 \pm 0.03$ | $-7.44 \pm 0.05$ | $-7.44 \pm 0.04$ | $-7.48 \pm 0.06$ |
| Large / Large | $-11.12 \pm 0.05$ | $\mathbf{-9.44 \pm 0.02}$ | $-9.67 \pm 0.05$ | $-10.37 \pm 0.02$ | $-9.66 \pm 0.04$ | $-9.68 \pm 0.05$ | $-9.70 \pm 0.05$ | $-9.69 \pm 0.06$ |
| Average | -8.41 | -7.92 | **-7.82** | -8.06 | -7.95 | -7.93 | -8.05 | -7.92 |

Table 14: **Unnormalized MSD Results**.

| | PID Controller | GRU | Transformer | PIDE | GPIDE | GPIDE-ES | GPIDE-ESS | GPIDE-Attn |
|---|---|---|---|---|---|---|---|---|
| Fixed / Fixed | $58.09 \pm 1.66$ | $93.18 \pm 1.46$ | $94.04 \pm 0.83$ | $\mathbf{100.00 \pm 0.27}$ | $93.18 \pm 1.20$ | $93.77 \pm 1.26$ | $96.20 \pm 1.22$ | $87.16 \pm 1.78$ |
| Fixed / Small | $36.41 \pm 5.36$ | $29.74 \pm 3.82$ | $\mathbf{64.96 \pm 1.38}$ | $54.90 \pm 0.73$ | $59.54 \pm 5.78$ | $54.35 \pm 1.35$ | $60.89 \pm 3.48$ | $51.84 \pm 3.38$ |
| Fixed / Large | $36.58 \pm 2.86$ | $0.00 \pm 3.42$ | $\mathbf{53.70 \pm 1.71}$ | $34.92 \pm 0.93$ | $29.55 \pm 2.32$ | $33.62 \pm 1.71$ | $0.60 \pm 6.25$ | $32.51 \pm 3.29$ |
| Small / Fixed | $46.87 \pm 5.88$ | $\mathbf{89.05 \pm 0.32}$ | $78.21 \pm 1.31$ | $75.81 \pm 0.79$ | $77.27 \pm 4.66$ | $81.41 \pm 1.20$ | $78.82 \pm 1.44$ | $79.64 \pm 1.80$ |
| Small / Small | $38.25 \pm 3.44$ | $\mathbf{100.00 \pm 0.98}$ | $83.49 \pm 3.07$ | $84.88 \pm 0.81$ | $87.78 \pm 5.31$ | $90.66 \pm 2.02$ | $83.97 \pm 2.40$ | $87.57 \pm 2.65$ |
| Small / Large | $43.52 \pm 2.82$ | $\mathbf{87.63 \pm 2.28}$ | $81.44 \pm 0.82$ | $53.21 \pm 1.31$ | $68.03 \pm 4.43$ | $68.09 \pm 4.10$ | $67.78 \pm 3.84$ | $79.57 \pm 4.71$ |
| Large / Fixed | $0.00 \pm 15.12$ | $\mathbf{63.36 \pm 1.17}$ | $45.01 \pm 3.18$ | $46.37 \pm 1.29$ | $46.68 \pm 3.06$ | $49.86 \pm 3.84$ | $48.52 \pm 3.69$ | $45.03 \pm 4.72$ |
| Large / Small | $0.00 \pm 15.75$ | $\mathbf{70.44 \pm 3.30}$ | $45.45 \pm 6.93$ | $45.73 \pm 2.47$ | $43.66 \pm 4.47$ | $44.71 \pm 6.45$ | $45.21 \pm 5.42$ | $39.64 \pm 7.82$ |
| Large / Large | $45.60 \pm 1.71$ | $\mathbf{100.00 \pm 0.61}$ | $92.60 \pm 1.49$ | $69.88 \pm 0.69$ | $93.03 \pm 1.27$ | $92.36 \pm 1.62$ | $91.67 \pm 1.68$ | $91.95 \pm 1.80$ |
| Average | 33.92 | 70.38 | **70.99** | 62.86 | 66.53 | 67.65 | 63.74 | 66.10 |

Table 15: **Normalized MSD Results**.

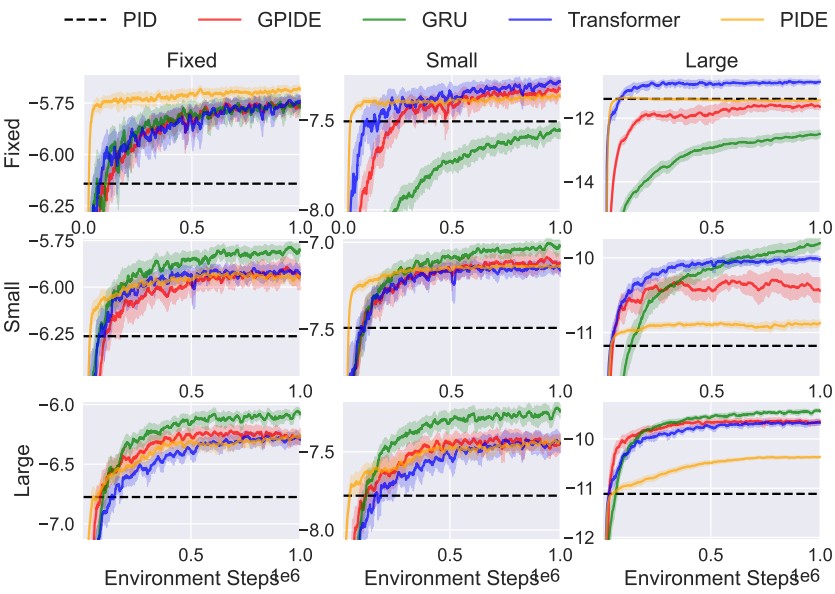

Figure 10: **MSD Performance Curves.** Each row corresponds to a training environment, and each column corresponds to a testing environment.

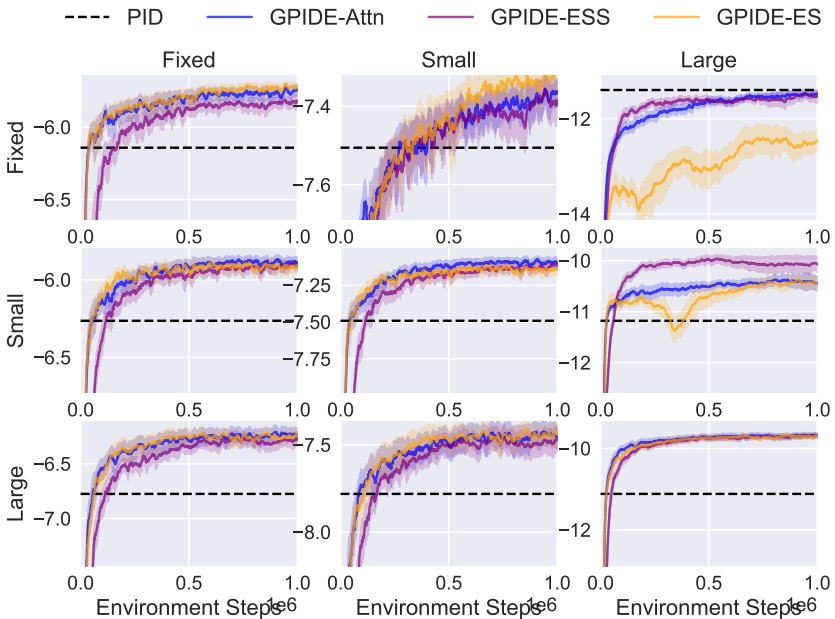

Figure 11: **MSD Performance Curve for Ablations.** Each row corresponds to a training environment, and each column corresponds to a testing environment.

| | PID Controller | GRU | Transformer | PIDE | GPIDE | GPIDE-ES | GPIDE-ESS | GPIDE-Attn |
|---|---|---|---|---|---|---|---|---|
| Fixed / Fixed | $-15.33 \pm 0.14$ | $-16.20 \pm 0.31$ | $-15.41 \pm 0.13$ | $\mathbf{-12.64 \pm 0.04}$ | $-13.49 \pm 0.22$ | $-13.92 \pm 0.09$ | $-13.35 \pm 0.05$ | $-16.77 \pm 0.13$ |
| Fixed / Small | $-21.29 \pm 0.29$ | $-25.21 \pm 0.32$ | $-21.37 \pm 0.16$ | $\mathbf{-18.58 \pm 0.05}$ | $-19.77 \pm 0.24$ | $-21.31 \pm 0.07$ | $-20.09 \pm 0.08$ | $-23.29 \pm 0.15$ |
| Fixed / Large | $-27.59 \pm 0.44$ | $-37.21 \pm 0.35$ | $-28.16 \pm 0.17$ | $\mathbf{-25.29 \pm 0.18}$ | $-27.54 \pm 0.33$ | $-31.14 \pm 0.13$ | $-28.14 \pm 0.11$ | $-31.84 \pm 0.71$ |
| Small / Fixed | $-18.15 \pm 0.91$ | $-17.75 \pm 0.42$ | $-15.86 \pm 0.11$ | $\mathbf{-13.43 \pm 0.09}$ | $-14.37 \pm 0.17$ | $-14.35 \pm 0.11$ | $-13.57 \pm 0.10$ | $-16.85 \pm 0.11$ |
| Small / Small | $-21.78 \pm 0.14$ | $-22.49 \pm 0.34$ | $-20.56 \pm 0.16$ | $-18.09 \pm 0.04$ | $-18.67 \pm 0.17$ | $-18.93 \pm 0.10$ | $\mathbf{-17.97 \pm 0.07}$ | $-21.77 \pm 0.10$ |
| Small / Large | $-26.57 \pm 0.22$ | $-31.27 \pm 0.36$ | $-26.04 \pm 0.24$ | $-23.82 \pm 0.13$ | $-23.65 \pm 0.20$ | $-23.66 \pm 0.10$ | $\mathbf{-22.72 \pm 0.08}$ | $-28.26 \pm 0.12$ |
| Large / Fixed | $-21.96 \pm 0.62$ | $-22.41 \pm 0.32$ | $-18.37 \pm 0.30$ | $\mathbf{-14.83 \pm 0.12}$ | $-15.75 \pm 0.14$ | $-16.79 \pm 0.04$ | $-15.23 \pm 0.12$ | $-18.89 \pm 0.28$ |
| Large / Small | $-22.30 \pm 0.44$ | $-26.63 \pm 0.39$ | $-22.00 \pm 0.24$ | $\mathbf{-19.46 \pm 0.08}$ | $-19.99 \pm 0.15$ | $-21.14 \pm 0.07$ | $-19.71 \pm 0.12$ | $-23.19 \pm 0.32$ |
| Large / Large | $-25.29 \pm 0.30$ | $-29.34 \pm 0.30$ | $-24.43 \pm 0.21$ | $-24.06 \pm 0.03$ | $-22.08 \pm 0.14$ | $-23.06 \pm 0.07$ | $\mathbf{-21.81 \pm 0.09}$ | $-25.32 \pm 0.19$ |
| Average | -22.25 | -25.39 | -21.36 | **-18.91** | -19.48 | -20.48 | -19.18 | -22.91 |

Table 16: **Unnormalized DMSD Results**.

| | PID Controller | GRU | Transformer | PIDE | GPIDE | GPIDE-ES | GPIDE-ESS | GPIDE-Attn |
|---|---|---|---|---|---|---|---|---|
| Fixed / Fixed | $72.45 \pm 1.44$ | $63.59 \pm 3.16$ | $71.62 \pm 1.30$ | $\mathbf{100.00 \pm 0.39}$ | $91.35 \pm 2.28$ | $86.93 \pm 0.89$ | $92.74 \pm 0.55$ | $57.75 \pm 1.31$ |
| Fixed / Small | $61.66 \pm 3.35$ | $16.43 \pm 3.75$ | $60.71 \pm 1.80$ | $\mathbf{93.01 \pm 0.60}$ | $79.26 \pm 2.81$ | $61.50 \pm 0.81$ | $75.51 \pm 0.97$ | $38.55 \pm 1.77$ |
| Fixed / Large | $62.47 \pm 2.86$ | $0.00 \pm 2.24$ | $58.78 \pm 1.11$ | $\mathbf{77.38 \pm 1.14}$ | $62.76 \pm 2.13$ | $39.41 \pm 0.83$ | $58.92 \pm 0.73$ | $34.84 \pm 4.61$ |
| Small / Fixed | $43.59 \pm 9.27$ | $47.76 \pm 4.25$ | $67.02 \pm 1.10$ | $\mathbf{91.92 \pm 0.90}$ | $82.32 \pm 1.72$ | $82.52 \pm 1.14$ | $90.46 \pm 0.99$ | $56.98 \pm 1.16$ |
| Small / Small | $56.04 \pm 1.57$ | $47.82 \pm 3.96$ | $70.07 \pm 1.88$ | $98.69 \pm 0.48$ | $91.94 \pm 2.00$ | $88.95 \pm 1.18$ | $\mathbf{100.00 \pm 0.78}$ | $56.17 \pm 1.11$ |
| Small / Large | $69.11 \pm 1.42$ | $38.57 \pm 2.33$ | $72.51 \pm 1.58$ | $86.96 \pm 0.82$ | $88.08 \pm 1.31$ | $87.99 \pm 0.64$ | $\mathbf{94.09 \pm 0.51}$ | $58.08 \pm 0.80$ |
| Large / Fixed | $4.64 \pm 6.34$ | $0.00 \pm 3.30$ | $41.37 \pm 3.09$ | $\mathbf{77.62 \pm 1.24}$ | $68.16 \pm 1.45$ | $57.60 \pm 0.36$ | $73.51 \pm 1.24$ | $36.06 \pm 2.85$ |
| Large / Small | $50.02 \pm 5.07$ | $0.00 \pm 4.56$ | $53.45 \pm 2.80$ | $\mathbf{82.77 \pm 0.98}$ | $76.66 \pm 1.75$ | $63.36 \pm 0.85$ | $79.93 \pm 1.43$ | $39.74 \pm 3.65$ |
| Large / Large | $77.38 \pm 1.93$ | $51.09 \pm 1.98$ | $82.96 \pm 1.38$ | $85.37 \pm 0.18$ | $98.23 \pm 0.90$ | $91.86 \pm 0.44$ | $\mathbf{100.00 \pm 0.56}$ | $77.22 \pm 1.21$ |
| Average | 55.26 | 29.47 | 64.28 | **88.19** | 82.08 | 73.35 | 85.02 | 50.60 |

Table 17: **Normalized DMSD Results**.

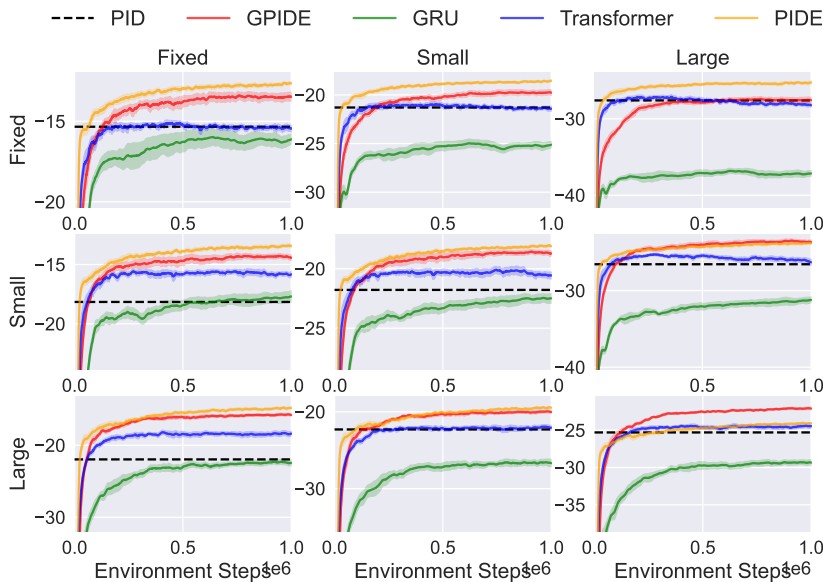

Figure 12: **DMSD Performance Curves.** Each row corresponds to a training environment, and each column corresponds to a testing environment.

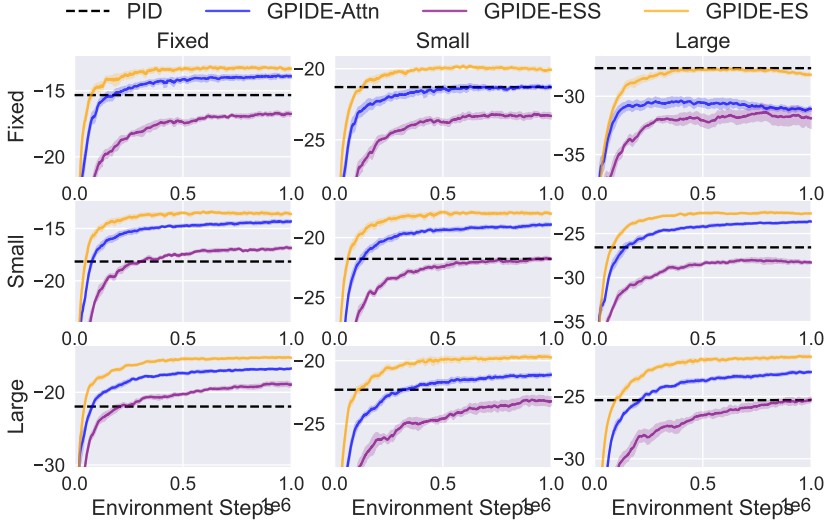

Figure 13: **DMSD Performance Curve for Ablations.** Each row corresponds to a training environment, and each column corresponds to a testing environment.

## G.3 Navigation Results

| | PID Controller | GRU | Transformer | PIDE | GPIDE | GPIDE-ES | GPIDE-ESS | GPIDE-Attn |
|---|---|---|---|---|---|---|---|---|
| Sim / Sim | $28.94 \pm 3.63$ | $96.76 \pm 0.15$ | $99.57 \pm 0.12$ | $98.33 \pm 0.06$ | $\mathbf{100.00 \pm 0.07}$ | $99.64 \pm 0.06$ | $99.66 \pm 0.06$ | $99.81 \pm 0.09$ |
| Sim / Real | $43.12 \pm 2.08$ | $0.00 \pm 3.94$ | $50.55 \pm 0.78$ | $\mathbf{68.34 \pm 0.57}$ | $62.16 \pm 0.89$ | $63.17 \pm 0.57$ | $59.21 \pm 1.15$ | $52.52 \pm 0.50$ |
| Real / Sim | $0.00 \pm 4.09$ | $57.49 \pm 1.17$ | $68.03 \pm 0.40$ | $59.54 \pm 0.85$ | $\mathbf{74.88 \pm 0.61}$ | $72.84 \pm 0.64$ | $74.75 \pm 0.68$ | $71.13 \pm 0.72$ |
| Real / Real | $67.28 \pm 2.05$ | $97.29 \pm 0.20$ | $99.20 \pm 0.14$ | $95.94 \pm 0.04$ | $\mathbf{100.00 \pm 0.21}$ | $99.19 \pm 0.09$ | $99.11 \pm 0.21$ | $99.67 \pm 0.17$ |
| Average | 34.83 | 62.89 | 79.34 | 80.54 | **84.26** | 83.71 | 83.18 | 80.78 |

Table 18: **Normalized Navigation Results**. Note that these results are after 1 million collected samples.

| | PID Controller | GRU | Transformer | PIDE | GPIDE | GPIDE-ES | GPIDE-ESS | GPIDE-Attn |
|---|---|---|---|---|---|---|---|---|
| Sim / Sim | $-17.23 \pm 0.18$ | $-13.82 \pm 0.01$ | $-13.68 \pm 0.01$ | $-13.74 \pm 0.00$ | $\mathbf{-13.65 \pm 0.00}$ | $-13.67 \pm 0.00$ | $-13.67 \pm 0.00$ | $-13.66 \pm 0.00$ |
| Sim / Real | $-23.87 \pm 0.29$ | $-29.85 \pm 0.55$ | $-22.84 \pm 0.11$ | $\mathbf{-20.37 \pm 0.08}$ | $-21.23 \pm 0.12$ | $-21.09 \pm 0.08$ | $-21.64 \pm 0.16$ | $-22.57 \pm 0.07$ |
| Real / Sim | $-18.69 \pm 0.21$ | $-15.79 \pm 0.06$ | $-15.26 \pm 0.02$ | $-15.69 \pm 0.04$ | $\mathbf{-14.92 \pm 0.03}$ | $-15.02 \pm 0.03$ | $-14.93 \pm 0.03$ | $-15.11 \pm 0.04$ |
| Real / Real | $-20.52 \pm 0.28$ | $-16.36 \pm 0.03$ | $-16.09 \pm 0.02$ | $-16.55 \pm 0.01$ | $\mathbf{-15.98 \pm 0.03}$ | $-16.09 \pm 0.01$ | $-16.11 \pm 0.03$ | $-16.03 \pm 0.02$ |
| Average | -20.08 | -18.96 | -16.97 | -16.59 | **-16.45** | -16.47 | -16.59 | -16.84 |

Table 19: **Unnormalized Navigation Results**. Note that these results are after 1 million collected samples.

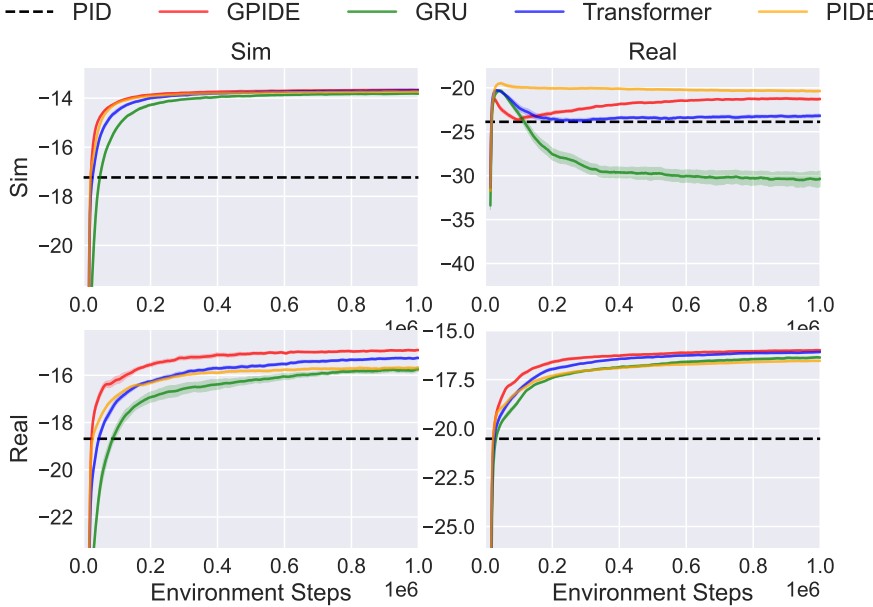

Figure 14: **Navigation Performance Curves.** Each row corresponds to a training environment, and each column corresponds to a testing environment. Note that these runs are only done for one million transitions.

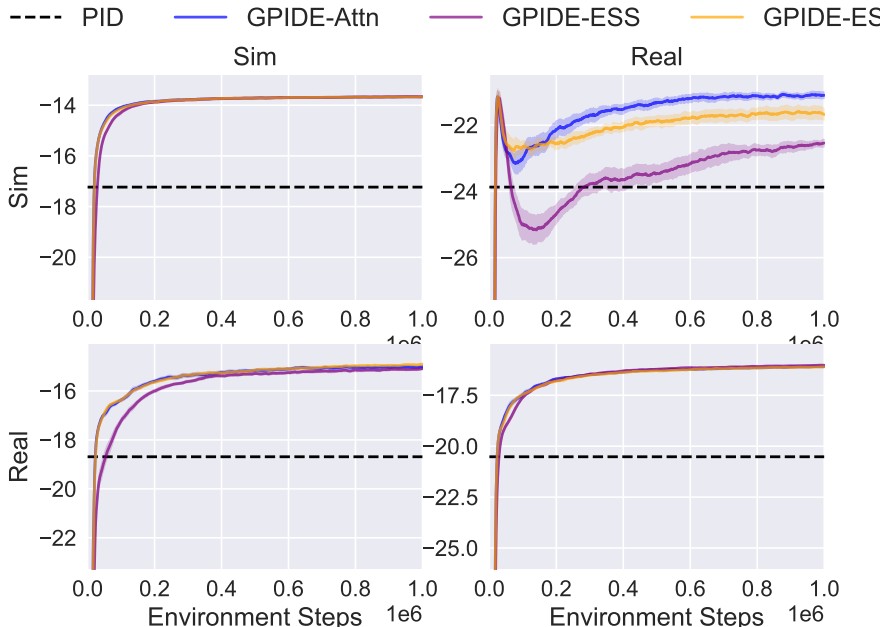

Figure 15: **Navigation Performance Curve for Ablations.** Each row corresponds to a training environment, and each column corresponds to a testing environment. Note that these runs are only done for one million transitions.

## G.4 Tokamak Control Results

|  | PID Controller | GRU | Transformer | PIDE | GPIDE | GPIDE-ES | GPIDE-ESS | GPIDE-Attn |
|---|---|---|---|---|---|---|---|---|
| Sim / Sim | $90.95 \pm 0.05$ | $\mathbf{100.00 \pm 0.03}$ | $99.63 \pm 0.03$ | $84.74 \pm 0.16$ | $99.75 \pm 0.06$ | $99.91 \pm 0.02$ | $99.90 \pm 0.02$ | $99.47 \pm 0.04$ |
| Sim / Real | $\mathbf{89.15 \pm 0.99}$ | $40.96 \pm 5.45$ | $40.05 \pm 11.91$ | $0.00 \pm 21.04$ | $55.21 \pm 4.44$ | $61.56 \pm 7.40$ | $65.65 \pm 5.66$ | $35.66 \pm 4.41$ |
| Real / Sim | $50.62 \pm 3.96$ | $36.33 \pm 3.61$ | $35.26 \pm 2.22$ | $0.00 \pm 3.48$ | $48.40 \pm 4.04$ | $52.62 \pm 1.38$ | $\mathbf{56.30 \pm 2.25}$ | $16.33 \pm 5.98$ |
| Real / Real | $98.45 \pm 0.77$ | $98.24 \pm 0.38$ | $98.74 \pm 0.29$ | $\mathbf{100.00 \pm 0.23}$ | $99.30 \pm 0.64$ | $98.39 \pm 0.33$ | $98.55 \pm 0.33$ | $98.27 \pm 0.37$ |
| Average | $\mathbf{82.29}$ | 68.88 | 68.42 | 46.18 | 75.67 | 78.12 | 80.10 | 62.43 |

Table 20: **Normalized $\beta_N$ Tracking Results**.

|  | PID Controller | GRU | Transformer | PIDE | GPIDE | GPIDE-ES | GPIDE-ESS | GPIDE-Attn |
|---|---|---|---|---|---|---|---|---|
| Sim / Sim | $-8.09 \pm 0.00$ | $\mathbf{-7.19 \pm 0.00}$ | $-7.22 \pm 0.00$ | $-8.71 \pm 0.02$ | $-7.21 \pm 0.01$ | $-7.19 \pm 0.00$ | $-7.20 \pm 0.00$ | $-7.24 \pm 0.00$ |
| Sim / Real | $\mathbf{-16.41 \pm 0.30}$ | $-31.21 \pm 1.67$ | $-31.49 \pm 3.66$ | $-43.78 \pm 6.46$ | $-26.83 \pm 1.36$ | $-24.88 \pm 2.27$ | $-23.63 \pm 1.74$ | $-32.83 \pm 1.35$ |
| Real / Sim | $-12.12 \pm 0.40$ | $-13.55 \pm 0.36$ | $-13.66 \pm 0.22$ | $-17.18 \pm 0.35$ | $-12.34 \pm 0.40$ | $-11.92 \pm 0.14$ | $\mathbf{-11.55 \pm 0.22}$ | $-15.55 \pm 0.60$ |
| Real / Real | $-13.56 \pm 0.23$ | $-13.62 \pm 0.12$ | $-13.47 \pm 0.09$ | $\mathbf{-13.08 \pm 0.07}$ | $-13.30 \pm 0.20$ | $-13.58 \pm 0.10$ | $-13.53 \pm 0.10$ | $-13.61 \pm 0.11$ |
| Average | $\mathbf{-12.55}$ | -16.39 | -16.46 | -20.69 | -14.92 | -14.39 | -13.98 | -17.31 |

Table 21: **Unnormalized $\beta_N$ Tracking Results**.

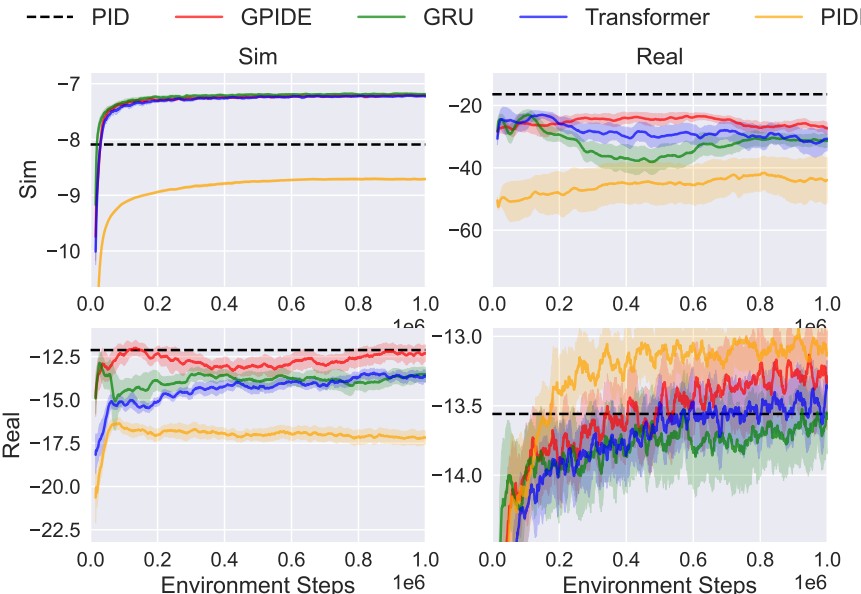

Figure 16: $\beta_N$ **Tracking Performance Curves.** Each row corresponds to a training environment, and each column corresponds to a testing environment.

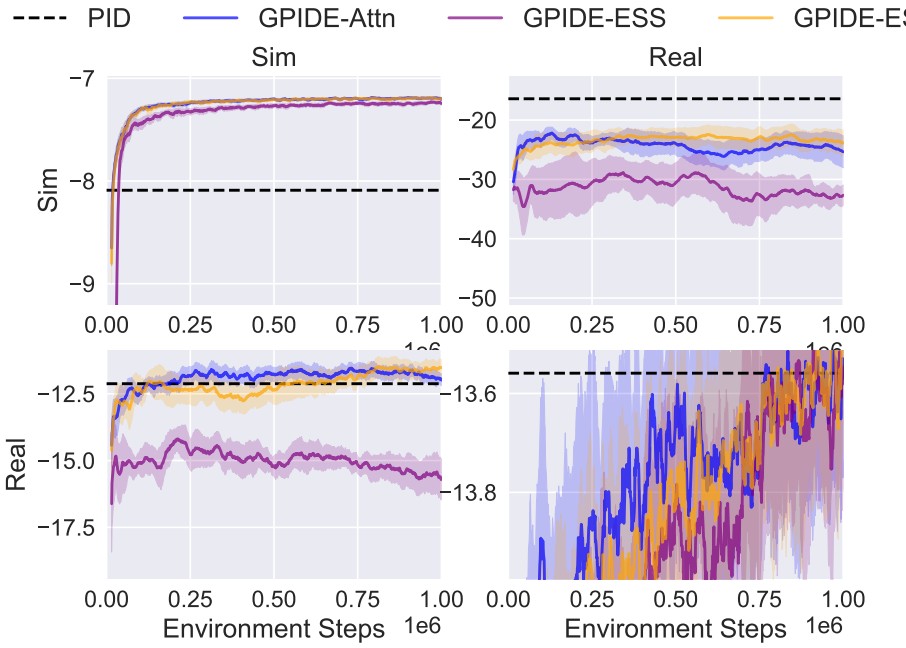

Figure 17: $\beta_N$ **Tracking Performance Curve for Ablations.** Each row corresponds to a training environment, and each column corresponds to a testing environment.

| | PID Controller | GRU | Transformer | PIDE | GPIDE | GPIDE-ES | GPIDE-ESS | GPIDE-Attn |
|---|---|---|---|---|---|---|---|---|
| Sim / Sim | $46.78 \pm 0.44$ | $99.50 \pm 0.12$ | $97.99 \pm 0.50$ | $82.96 \pm 0.27$ | $\mathbf{100.00 \pm 0.15}$ | $99.64 \pm 0.19$ | $99.97 \pm 0.12$ | $96.18 \pm 1.35$ |
| Sim / Real | $\mathbf{83.48 \pm 2.63}$ | $39.65 \pm 5.83$ | $33.22 \pm 0.69$ | $0.00 \pm 8.87$ | $50.86 \pm 1.92$ | $54.36 \pm 2.07$ | $52.56 \pm 2.44$ | $42.51 \pm 2.97$ |
| Real / Sim | $0.00 \pm 8.79$ | $21.31 \pm 2.45$ | $7.23 \pm 3.86$ | $22.49 \pm 1.84$ | $19.02 \pm 3.88$ | $\mathbf{22.70 \pm 4.42}$ | $5.20 \pm 20.06$ | $15.35 \pm 8.29$ |
| Real / Real | $91.76 \pm 0.84$ | $98.07 \pm 0.52$ | $96.05 \pm 0.31$ | $97.94 \pm 0.23$ | $99.73 \pm 0.46$ | $97.62 \pm 0.46$ | $\mathbf{100.00 \pm 0.28}$ | $96.33 \pm 0.47$ |
| Average | $55.51$ | $64.63$ | $58.62$ | $50.85$ | $67.40$ | $\mathbf{68.58}$ | $64.43$ | $62.59$ |

Table 22: **Normalized $\beta_N$-Rotation Tracking Results**.

| | PID Controller | GRU | Transformer | PIDE | GPIDE | GPIDE-ES | GPIDE-ESS | GPIDE-Attn |
|---|---|---|---|---|---|---|---|---|
| Sim / Sim | $-27.56 \pm 0.08$ | $-18.53 \pm 0.02$ | $-18.79 \pm 0.09$ | $-21.36 \pm 0.05$ | $\mathbf{-18.45 \pm 0.03}$ | $-18.51 \pm 0.03$ | $-18.45 \pm 0.02$ | $-19.10 \pm 0.23$ |
| Sim / Real | $\mathbf{-30.08 \pm 0.95}$ | $-45.91 \pm 2.10$ | $-48.23 \pm 0.25$ | $-60.23 \pm 3.20$ | $-41.86 \pm 0.69$ | $-40.60 \pm 0.75$ | $-41.25 \pm 0.88$ | $-44.88 \pm 1.07$ |
| Real / Sim | $-35.57 \pm 1.50$ | $-31.92 \pm 0.42$ | $-34.33 \pm 0.66$ | $-31.72 \pm 0.32$ | $-32.31 \pm 0.66$ | $\mathbf{-31.68 \pm 0.76}$ | $-34.68 \pm 3.43$ | $-32.94 \pm 1.42$ |
| Real / Real | $-27.09 \pm 0.30$ | $-24.81 \pm 0.19$ | $-25.54 \pm 0.11$ | $-24.86 \pm 0.08$ | $-24.21 \pm 0.16$ | $-24.98 \pm 0.17$ | $\mathbf{-24.12 \pm 0.10}$ | $-25.44 \pm 0.17$ |
| Average | $-30.08$ | $-30.29$ | $-31.72$ | $-34.54$ | $-29.21$ | $\mathbf{-28.94}$ | $-29.62$ | $-30.59$ |

Table 23: **Unnormalized $\beta_N$-Rotation Tracking Results**.

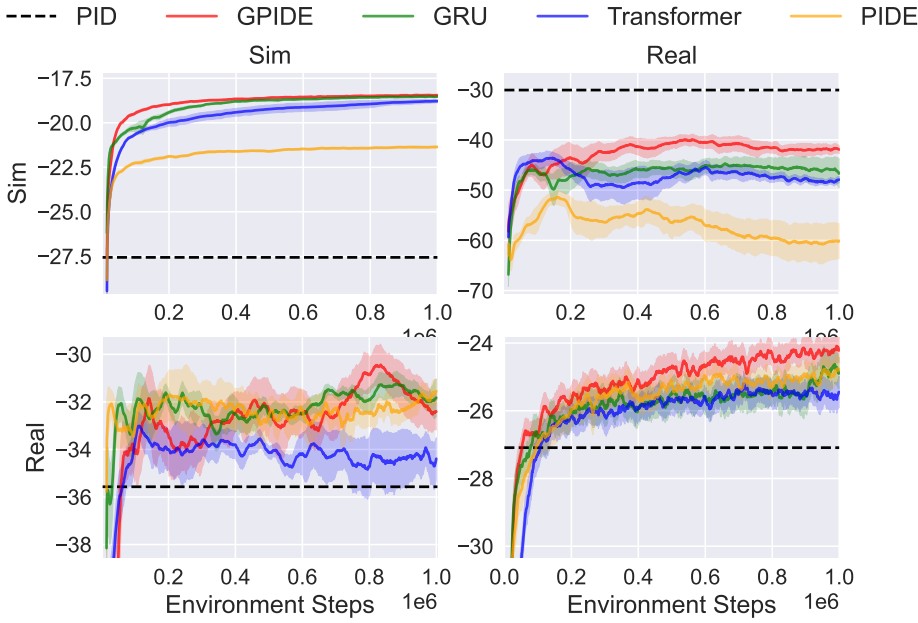

Figure 18: $\beta_N$-**Rotation Tracking Performance Curves.** Each row corresponds to a training environment, and each column corresponds to a testing environment.

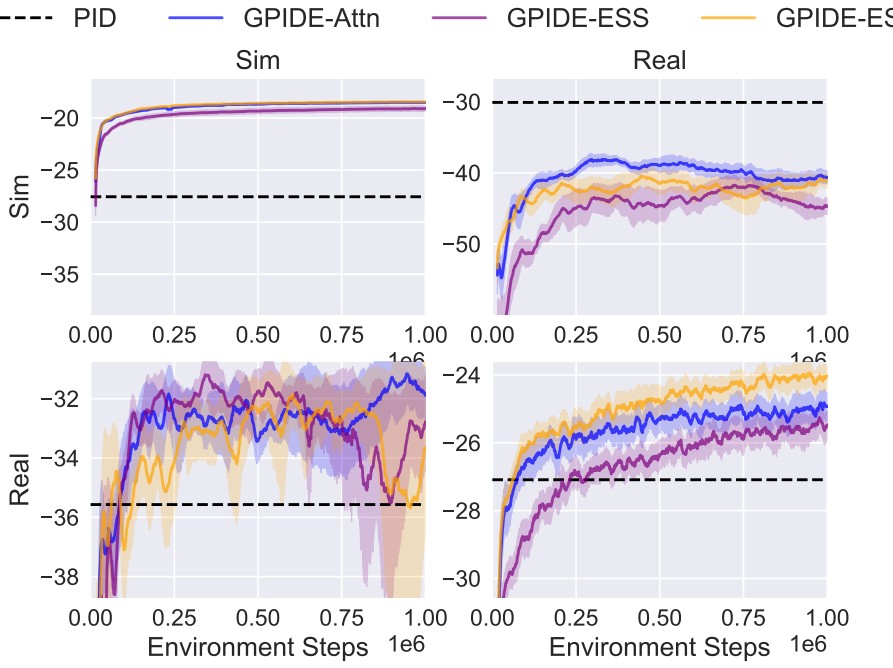

Figure 19: $\beta_N$-**Rotation Tracking Performance Curve for Ablations.** Each row corresponds to a training environment, and each column corresponds to a testing environment.

## G.5 PyBullet Results

For these results, SAC encodes observations, actions and rewards. TD3 encodes observations and actions since it is the best performing on average.

| | PPO-GRU | A2C-GRU | SAC-LSTM | TD3-GRU | VRM | SAC-Transformer | SAC-GPIDE | SAC-GPIDE-ES | SAC-GPIDE-ESS | SAC-GPIDE-Attn |
|---|---|---|---|---|---|---|---|---|---|---|
| HalfCheetah-P | $27.09 \pm 7.85$ | $-22.00 \pm 5.13$ | $77.06 \pm 7.96$ | $\mathbf{85.80 \pm 5.15}$ | $-107.00 \pm 1.39$ | $37.00 \pm 9.97$ | $82.63 \pm 3.46$ | $85.45 \pm 4.83$ | $71.08 \pm 6.95$ | $77.63 \pm 4.60$ |
| Hopper-P | $49.00 \pm 5.22$ | $-2.29 \pm 3.33$ | $64.36 \pm 9.94$ | $84.63 \pm 8.33$ | $3.53 \pm 1.63$ | $59.54 \pm 19.64$ | $93.27 \pm 13.56$ | $111.48 \pm 4.20$ | $\mathbf{113.95 \pm 4.67}$ | $104.87 \pm 8.76$ |
| Walker-P | $1.67 \pm 4.39$ | $-10.48 \pm 1.73$ | $40.92 \pm 15.56$ | $29.08 \pm 9.67$ | $-3.89 \pm 1.25$ | $24.89 \pm 14.80$ | $\mathbf{96.61 \pm 1.60}$ | $76.58 \pm 5.47$ | $94.58 \pm 11.00$ | $71.36 \pm 6.57$ |
| Ant-P | $39.48 \pm 3.74$ | $-13.06 \pm 6.52$ | $60.97 \pm 3.54$ | $-36.36 \pm 3.35$ | $-36.39 \pm 0.17$ | $-10.57 \pm 2.34$ | $\mathbf{66.66 \pm 2.94}$ | $64.73 \pm 3.82$ | $57.78 \pm 3.78$ | $63.19 \pm 5.32$ |
| HalfCheetah-V | $19.68 \pm 11.71$ | $-50.13 \pm 9.50$ | $18.54 \pm 33.09$ | $\mathbf{59.03 \pm 2.88}$ | $-80.49 \pm 2.97$ | $-41.31 \pm 26.15$ | $20.39 \pm 29.60$ | $51.03 \pm 13.93$ | $53.14 \pm 5.86$ | $-54.70 \pm 19.89$ |
| Hopper-V | $13.86 \pm 4.80$ | $-0.60 \pm 3.33$ | $16.26 \pm 12.44$ | $57.43 \pm 8.63$ | $10.08 \pm 3.51$ | $0.28 \pm 8.49$ | $\mathbf{90.98 \pm 4.28}$ | $72.63 \pm 19.28$ | $90.09 \pm 2.50$ | $30.73 \pm 1.60$ |
| Walker-V | $8.12 \pm 5.43$ | $-8.02 \pm 0.57$ | $-1.57 \pm 1.88$ | $-4.63 \pm 1.30$ | $-1.80 \pm 0.70$ | $-8.21 \pm 1.31$ | $36.90 \pm 16.59$ | $\mathbf{68.30 \pm 4.33}$ | $67.54 \pm 3.60$ | $14.85 \pm 11.26$ |
| Ant-V | $1.43 \pm 3.26$ | $-13.67 \pm 1.83$ | $-16.95 \pm 1.29$ | $17.03 \pm 6.55$ | $-13.41 \pm 0.12$ | $0.81 \pm 1.31$ | $\mathbf{18.03 \pm 5.10}$ | $4.56 \pm 5.20$ | $12.85 \pm 1.67$ | $-1.84 \pm 5.76$ |
| Average | 20.04 | -15.03 | 32.45 | 36.50 | -28.67 | 7.80 | 63.18 | 66.84 | **70.13** | 38.26 |

Table 24: **Normalized PyBullet Scores**.

| | PPO-GRU | A2C-GRU | SAC-LSTM | TD3-GRU | VRM | SAC-Transformer | SAC-GPIDE | SAC-GPIDE-ES | SAC-GPIDE-ESS | SAC-GPIDE-Attn |
|---|---|---|---|---|---|---|---|---|---|---|
| HalfCheetah-P | $1445.81 \pm 166.79$ | $403.35 \pm 108.97$ | $2506.88 \pm 168.93$ | $\mathbf{2692.53 \pm 109.43}$ | $-1401.67 \pm 29.62$ | $1656.13 \pm 211.75$ | $2625.13 \pm 73.49$ | $2684.98 \pm 102.57$ | $2379.79 \pm 147.67$ | $2519.06 \pm 97.72$ |
| Hopper-P | $1436.43 \pm 102.09$ | $433.19 \pm 65.09$ | $1736.81 \pm 194.51$ | $2133.42 \pm 162.93$ | $546.93 \pm 31.81$ | $1642.63 \pm 384.10$ | $2302.31 \pm 265.21$ | $2658.48 \pm 82.18$ | $\mathbf{2706.81 \pm 91.39}$ | $2529.31 \pm 171.41$ |
| Walker-P | $501.06 \pm 76.99$ | $288.10 \pm 30.39$ | $1189.28 \pm 272.77$ | $981.63 \pm 169.46$ | $403.60 \pm 21.85$ | $908.17 \pm 259.52$ | $\mathbf{2165.52 \pm 28.10}$ | $1814.40 \pm 95.91$ | $2129.91 \pm 192.87$ | $1722.81 \pm 115.22$ |
| Ant-P | $2025.52 \pm 84.58$ | $837.57 \pm 147.53$ | $2511.54 \pm 80.13$ | $310.72 \pm 75.68$ | $310.24 \pm 3.83$ | $893.84 \pm 52.83$ | $\mathbf{2640.16 \pm 66.46}$ | $2596.63 \pm 86.26$ | $2439.48 \pm 85.37$ | $2561.67 \pm 120.21$ |
| HalfCheetah-V | $1005.13 \pm 289.84$ | $-723.40 \pm 235.29$ | $977.02 \pm 819.24$ | $1979.56 \pm 71.40$ | $-1475.15 \pm 73.42$ | $-505.00 \pm 647.43$ | $1022.93 \pm 732.93$ | $1781.36 \pm 344.95$ | $1833.60 \pm 145.14$ | $-836.47 \pm 492.45$ |
| Hopper-V | $534.05 \pm 105.85$ | $215.22 \pm 73.48$ | $587.10 \pm 274.42$ | $1495.11 \pm 190.42$ | $450.77 \pm 77.35$ | $234.49 \pm 187.36$ | $\mathbf{2235.02 \pm 94.45}$ | $1830.26 \pm 425.16$ | $2215.47 \pm 55.16$ | $906.05 \pm 35.29$ |
| Walker-V | $377.80 \pm 109.11$ | $53.25 \pm 11.45$ | $182.97 \pm 37.89$ | $121.44 \pm 26.14$ | $178.28 \pm 14.09$ | $49.32 \pm 26.43$ | $956.43 \pm 333.46$ | $\mathbf{1587.56 \pm 87.15}$ | $1572.41 \pm 72.46$ | $513.07 \pm 226.34$ |
| Ant-V | $684.36 \pm 89.48$ | $269.32 \pm 50.35$ | $178.98 \pm 35.57$ | $1113.19 \pm 179.93$ | $276.33 \pm 3.18$ | $667.20 \pm 35.98$ | $\mathbf{1140.73 \pm 140.22}$ | $770.51 \pm 143.02$ | $998.35 \pm 46.04$ | $594.54 \pm 158.48$ |
| Average | 1001.27 | 222.08 | 1233.82 | 1353.45 | -88.84 | 693.35 | 1886.03 | 1965.52 | **2034.48** | 1313.76 |

Table 25: **Unnormalized PyBullet Scores**.

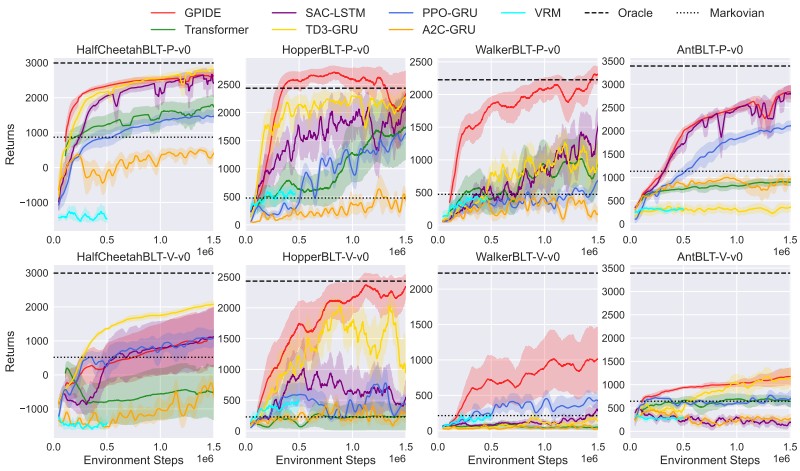

Figure 20: **PyBullet Performance Curves.**

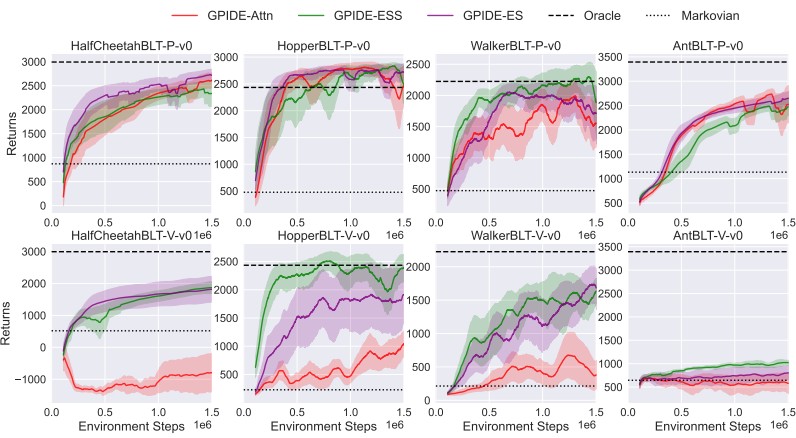

Figure 21: **PyBullet Performance Curve for Ablations.**

Interestingly, we found that GPIDE policies often outperform the oracle policy on Hopper-P. While the oracle performance here was taken from Ni et al. [50], we confirmed this also happens with our own implementation of an oracle policy. We hypothesize that this may be due to the fact the GPIDE policy gets to see actions and rewards and the oracle does not.

## G.6 Attention Scheme Visualizations

We generate the attention visualizations (as seen in Figure 4) by doing a handful of rollouts with a GPIDE policy using only attention heads. During this rollout we collect all of the weighting schemes, i.e. softmax $\left( \frac{q_{1:t} k_{1:t}^T}{\sqrt{D}} \right)$, generated throughout the rollouts and average them together. Below, we show additional attention visualizations. In all figures, each plot shows one of the different six heads. For each of these, the policies were evaluated on the same version of the environment they were trained on.

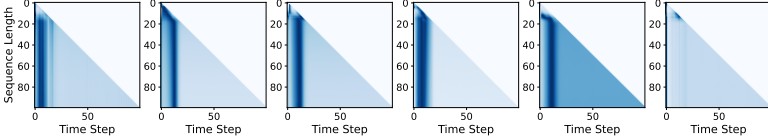

Figure 22: **MSD-Fixed Attention**.

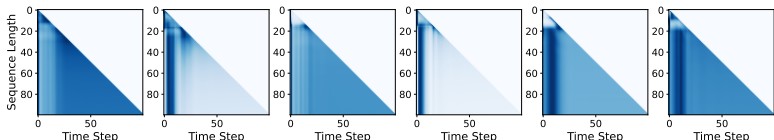

Figure 23: **MSD-Small Attention**.

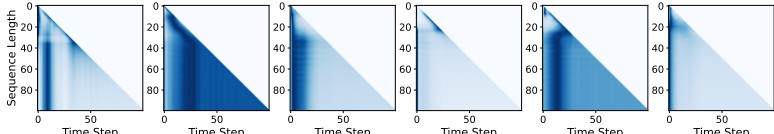

Figure 24: **MSD-Large Attention**.

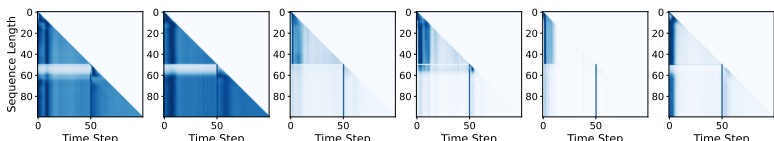

Figure 25: **DMSD-Fixed Attention**.

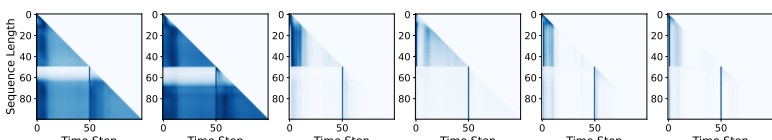

Figure 26: **DMSD-Small Attention**.

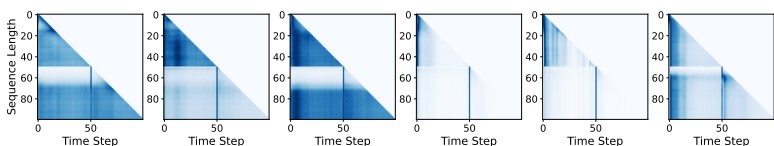

Figure 27: **DMSD-Large Attention**.

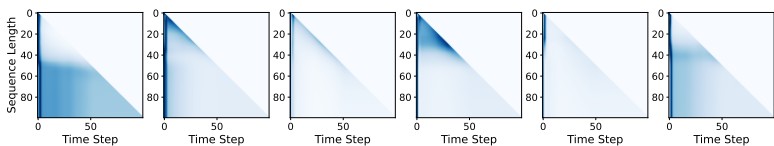

Figure 28: **Navigation No Friction Attention**.

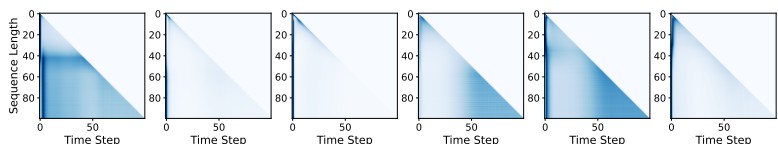

Figure 29: **Navigation Friction Attention**.

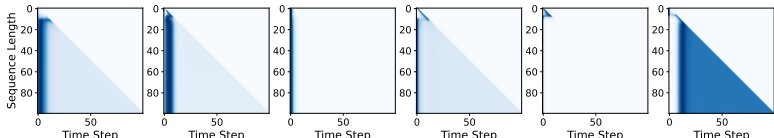

Figure 30: $\beta_N$ **Tracking Sim Attention**.

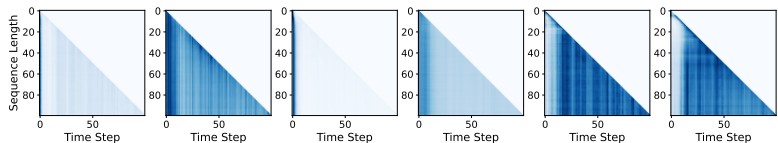

Figure 31: $\beta_N$ **Tracking Rotation Attention**.

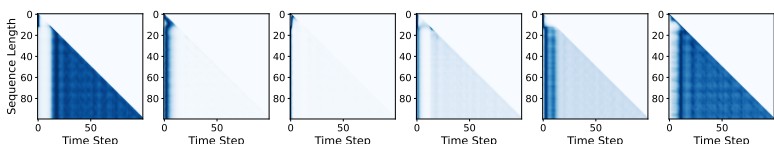

Figure 32: $\beta_N$**-Rotation Tracking Sim Attention**.

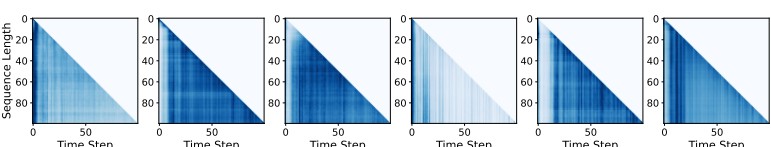

Figure 33: $\beta_N$**-Rotation Tracking Rotation Attention**.

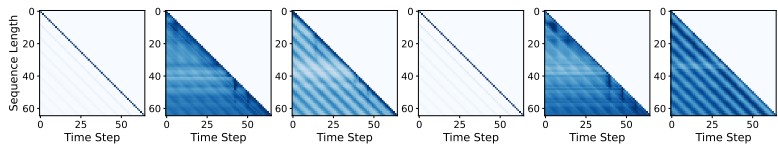

Figure 34: **HalfCheetah-P Attention**.

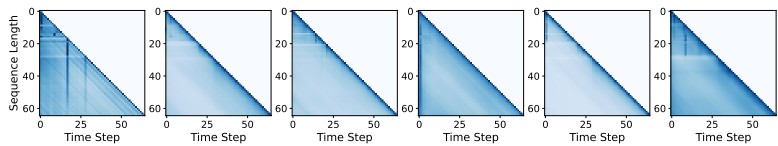

Figure 35: **HalfCheetah-V Attention**.

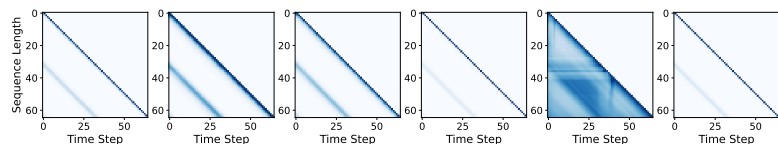

Figure 36: **Hopper-P Attention**.

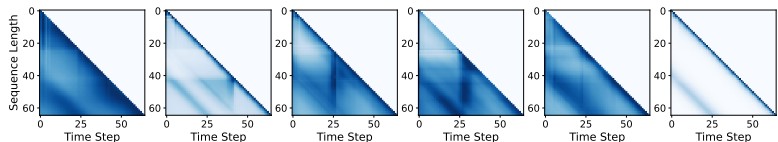

Figure 37: **Hopper-V Attention**.

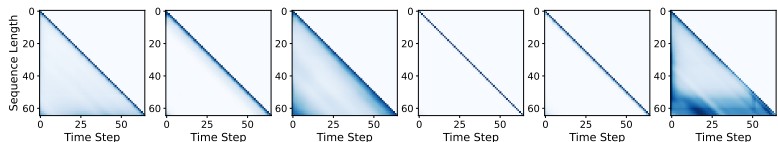

Figure 38: **Walker-P Attention**.

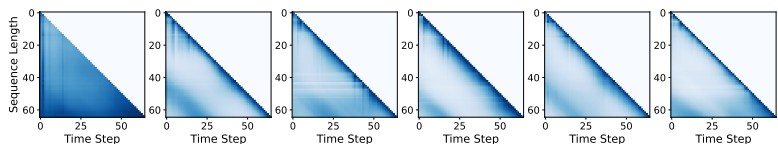

Figure 39: **Walker-V Attention**.

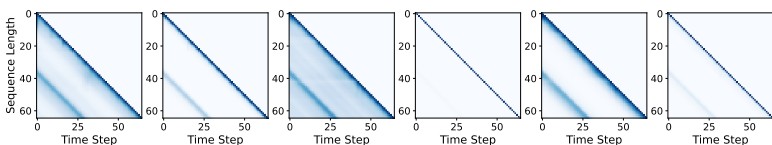

Figure 40: **Ant-P Attention**.

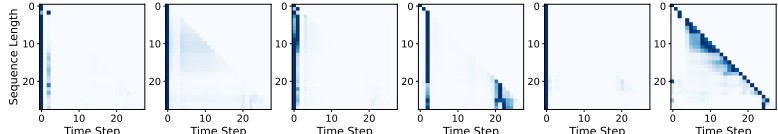

Figure 41: **Ant-V Attention**. Note that total path length is less than 64 here since the agent falls down pretty fast.

