# OpenReview forum: "PID-Inspired Inductive Biases for Deep Reinforcement Learning in Partially Observable Control Tasks"
_NeurIPS.cc/2023/Conference — NeurIPS 2023 poster_

### Official Review · Reviewer_RWrF · 2023-07-03

**Soundness:** 2 fair
**Presentation:** 3 good
**Contribution:** 3 good
**Rating:** 6
**Confidence:** 4

**Summary:**

The paper proposes two history encoders for RL in partially observable control tasks. The history encoders are designed with PID-inspired inductive bias. Specifically, the inductive bias comes from lifting the observation into features consisting of summation and difference of observation over the historical horizon, which are analogous to the integral and derivative of tracking errors in PID. It was shown that the proposed history encoders, especially the generalized PID encoder, could achieve better performance and robustness than GRU and transformers in tracking and locomotion tasks when integrated with RL algorithms.

**Strengths:**

I particularly appreciate the new perspective presented by the authors in combining conventional control techniques with reinforcement learning. The idea of incorporating PID architecture into history encoders is interesting and novel to the best of my knowledge. The advantages of the generalized PID encoders over GRU and transformers, especially in the locomotion tasks, validate that the PID-inspired inductive bias is indeed effective in some control tasks. This paper could inspire the RL community to discover more powerful and generalizable architectures with insights from control theory and conventional control techniques.

**Weaknesses:**

1. In lines 102-104, the authors stated:
>
> In the case of MIMO tracking problems, where there are M signals with M corresponding actuators, one can control the system with M separate PID controllers. However, this assumes there is a clear breakdown of which actuator influences which signal.
>
Decomposing a MIMO system into multiple SISO systems is indeed a practical way to synthesize MIMO PID controllers. However, there exist methods to directly synthesize PID controllers for MIMO systems without decomposition (e.g., [1]). While I am okay with the authors only comparing the proposed methods with a decomposed PID controller, the authors should appropriately assert the capacity of PID controllers based on the state-of-the-art literature.

2. The related work section should include a review of the literature on the combination of conventional control techniques with RL. For instance, there have been extensive efforts in tuning PID or learning an adaptive PID weight schedule with RL, which could be considered a special case of the control policy with the proposed PID encoder. I understand the paper is mainly targeting audiences from the RL community, but I think it is necessary to place the proposed method in the control literature as well.

3. While I am generally impressed by the experimental results, I am concerned about the failure of SAC-GPIDE in the HalfCheetah-V environment. It makes me wonder whether the PID-based inductive bias could limit the expressiveness of the policy network in some control tasks. For instance, I would doubt if the proposed method could still work when image-based observation is used. The authors do not provide any explanation or hypothesis for the failure in HalfCheetah-V. I would suggest providing more information and insights regarding it during the rebuttal and in the updated version of the paper.

[1] Boyd, Stephen, Martin Hast, and Karl Johan Åström. "MIMO PID tuning via iterated LMI restriction." International Journal of Robust and Nonlinear Control 26, no. 8 (2016): 1718-1731.


**Questions:**

1. Please see the three issues listed in the Weaknesses section. I would have a more positive impression of the paper if the authors could properly address these three issues during the rebuttal.

2. In addition, I want to ask the authors to clarify the role of attention heads in GPIDE. In the ablation study, the authors stated:
>
> It appears that the attention scheme for HalfCheetah is simply a poor reproduction of exponential smoothing, making it redundant and suboptimal. In fact, we found this phenomenon to be true across all attention heads and PyBullet tasks. We believe that the periodicity that appears here is due to the oscillatory nature of the problem and lack of positional encoding (although we found including positional encoding degrades performance).
>
If that is the case, does it mean that the attention heads could be simply replaced by exponential smoothing heads? It seems to be supported by the ablation study results (i.e., ES+SUM has similar or better performance than GPIDE with multiple types of heads). In lines 227-230, the authors said:
>
>This choice was not optimized but rather was picked so that all types of heads were included and so that GPIDE has roughly the same amount of parameters as our GRU baseline.
>
I don't think it is a good justification for the experimental design. The authors should attempt to find an optimized configuration so that the audience can have a better sense of the necessity of each proposed head, especially the attention head.


**Limitations:**

The authors acknowledged the limitation of the proposed methods, which is the proposed PID-inspired inductive bias might not apply to all control tasks. While I personally doubt if the proposed method can ever be extended to systems with image-based observations, I do not think the authors are obligated to justify the feasibility of extension to visual control tasks in this paper. However, as suggested above, I do think the authors should provide more information and insights on the failure of GPIDE in Cheetah-V.

---

> ### Author Rebuttal · Authors · 2023-08-09
>
> Thank you for your time and your thoughtful review! We are glad to hear that you were generally impressed with the experimental results. For the failure of HalfCheetah-V, we agree that the current iteration of the paper could benefit from more information. Please see the global response for what we plan to add.
>
> We also agree we need to do a better job at appropriately framing the capacity of PID based on state-of-the-art literature. On top of this, while our paper is about how PID can help RL, we agree that we do need to reference literature on how RL can help PID to give readers the full picture. In addition to the works recommended by reviewer EZtX, we also plan on referencing the works shown at the end of this response in our discussion.
>
> For your second question, although there are a few cases where attention may be helpful, generally it seems that the best performing variant of GPIDE does not have attention. We thought the version of the paper that would be most coherent to readers is to first run experiments with a “complete” GPIDE. That is, a version of the method that, although possibly suboptimal, uses all of the  heads discussed in the Methodology section. Through the ablation section we hoped to give readers insights into the importance of each of the proposed heads and hone in on a more optimal version of GPIDE.
>
> If you feel that we have adequately addressed your concerns, we would respectfully ask you to consider raising your score. Thank you!
>
> Lawrence, N. P., Stewart, G. E., Loewen, P. D., Forbes, M. G., Backstrom, J. U., & Gopaluni, R. B. (2020). Reinforcement learning based design of linear fixed structure controllers. IFAC-PapersOnLine, 53(2), 230-235.
>
> Guan, Z., & Yamamoto, T. (2021). Design of a Reinforcement Learning PID controller. IEEJ Transactions on Electrical and Electronic Engineering, 16(10), 1354-1360.
>
> Jesawada, H., Yerudkar, A., Del Vecchio, C., & Singh, N. (2022). A Model-Based Reinforcement Learning Approach for PID Design. arXiv preprint arXiv:2206.03567.

---

> > ### Comment · Reviewer_RWrF · 2023-08-16
> >
> > Thanks to the authors for the response! After reading the response and other reviewers' comments, I would like to keep my current score.

---

### Official Review · Reviewer_64AN · 2023-07-03

**Soundness:** 3 good
**Presentation:** 3 good
**Contribution:** 3 good
**Rating:** 6
**Confidence:** 3

**Summary:**

The paper proposes a new, simplified architecture for reinforcement learning in the partially observable setting inspired by PID control. Through experiments on a number of tracking problems, and some locomotion environments, strong performance is obtained.

**Strengths:**

- The exposition of the argument is very clear, even for a reader somewhat unfamiliar with some aspects.
- Simplification of complex (and computationally expensive) recurrent architectures is a worthwhile endeavor.
- Ablations are informative, and come with demonstrations of issues (Figure 4).

**Weaknesses:**

- The largest weakness is in the evaluation. Most of the evaluation is devoted to low-dimensional tracking problems. The choice of state-based locomotion with only positions or velocities observable also seems a little artificial when there are many tasks that have existing sources of partial observability (long-horizon, first-person viewpoints).
- One possibly beneficial use of an alternative state summary like GPIDE would be computational savings, but there isn't any measurement of that in the paper.

**Questions:**

See the final weakness, it would be nice to see computational cost measured.

**Limitations:**

Limitations are discussed.

---

> ### Author Rebuttal · Authors · 2023-08-09
>
> Thank you for your time and your review! For your note on computational savings, please refer to the global response.
>
> While it is true the majority of the environment variants (10 out of the 18) are lower-dimensional tracking problems (either 3 or 6 dimensions), we do not believe that they are any less valuable than the high dimensional ones. It would be one thing if all baselines performed optimally on the lower dimensional experiments, but we find that is not the case and that pre-existing (and widely used) methods can be very suboptimal. How can we expect a method to perform optimally on high-dimensional tasks if it cannot on basic low-dimensional ones?
>
> Lastly, while we agree there are many different types of partial observability that occur in real-world tasks, partial observability of positions or velocities is a common benchmark in the literature (Han et al 2019, Meng et al 2021, Yang and Nguyen 2021, Ni et al 2022).

---

> > ### Comment · Reviewer_64AN · 2023-08-14
> >
> > Thanks for your response, a couple of specific things below
> >
> > - **Computational savings**: I had a look at appendix E, and the details are only for training. While it's helpful, it would also be nice to get a sense of the inference costs for using GPIDE instead of baselines. For example, does it lead to a qualitative difference that lets some tasks run in real-time? This would be a significant win over baselines and I think is worth exposing.
> > - **Low vs. high dimensional environments**: I'm not totally convinced by this line of argument. I believe the goal is to do research on interesting tasks. The most obvious existing case of partial-observability to me is something like a first-person viewpoint. I would also take strategic games that have fog of war (Stratego, Starcraft, etc.) as other good examples. Another great example is minesweeper. While some of these domains are very hard for existing algorithms, others seem to be quite natural and have been used by many papers in different subfields (first-person navigation in particular). Such domains seem to be much closer to the eventual deployment scenario of the paper's method than masked tracking problems, though I would be convinced by a good argument as to the deployment uses of tracking. I do acknowledge that I am not familiar with the literature, and using standard domains is a good idea.

---

> > > ### Author Response · Authors · 2023-08-15
> > >
> > > Thank you for your reply! Your point about the test-time execution point is a good one that we had not considered during the writing of the paper, and we now realize that Appendix E does not fully shed light on this. As noted in our global response, we speculate that GPIDE without attention should be faster than a policy that uses GRU or LSTM; however, we cannot say for certain right now how much faster. We will work on creating a more efficient version of our code that assumes no attention is used to test this.
> > >
> > > We agree with you that research should be grounded in real, interesting applications. While the “fog-of-war” type of partial observability is interesting, we want to emphasize that there are many other types of partial observability worth studying, especially in robotic applications. For example, partial observability can come in the form of unknown system parameters (which are present in all of our tracking experiments) or unmeasured signals. While masking positions and velocities is one form of the latter that appears frequently in the literature, our tokamak control experiment has realistic assumptions on which states of the plasma can be measured in real time. Tokmak control using RL has gained attention from the community lately (e.g. see “Magnetic control of tokamak plasmas through deep reinforcement learning” from Degrave et al. which also considers a tracking problem), and this experiment highlights the partial observability challenges that this application faces.

---

> > > > ### Comment · Reviewer_64AN · 2023-08-15
> > > >
> > > > I agree that the justification for Tokamak control seems stronger, but it is only a small portion of the experiments. I can also think of a task like landing rockets SpaceX-style, where precise measurements are not possible for cheap hardware, yet, for example, weight affects the control signal to be generated. This might be a bit more satisfying and natural than the given toy domains.

---

### Official Review · Reviewer_QQYm · 2023-07-23

**Soundness:** 4 excellent
**Presentation:** 4 excellent
**Contribution:** 3 good
**Rating:** 7
**Confidence:** 4

**Summary:**

This paper considers history encoding for deep RL POMDP control problems through a PID-inspired lens. Specifically, authors introduce PIDE, a method of directly using PID control which extends to multiple-input multiple-output problems, and GPIDE, a PID-inspired encoder architecture. GPIDE consists of a series of heads that takes as input model observations and past history, and internally computes differences between observations (similar to the PID D-term) and aggregation/summation over all past timesteps (similar to the PID I-term).

The method is then evaluated on tasks including mass-spring-damper, navigation, tolemak control, and pyBullet locomotion. Across these experiments, the method is compared against baselines of direct PID control, GRU, transformer, and several RL algorithms. GPIDE is found to have strong performance across both simple and complex tasks and be robust to domain transfer scenarios. Further ablation studies provide insight into design choices for GPIDE heads, visualize attention schemes, and evaluate a GRU+VIB baseline.


**Strengths:**

#### Originality

The ideas of the paper are generally original, building from prior deep RL work and focusing on the design of the feature encoder, especially designing a novel history encoding architecture which draws from PID controllers. The pieces of the architecture draw from general temporal aggregation principles and transformers.

#### Quality

The paper is generally high-quality. The experiments compare against relevant baselines and prior work, are run with multiple seeds, and the training / setup / tasks are clearly defined. The code is provided and looks cleanly written from initial observation, though I haven’t run it to verify. The ablation studies answer my initial questions about the architecture design, especially in how important the attention heads are.

#### Clarity

The manuscript is well written and clearly conveys the derivation of the method from PID controllers and explains the architecture and design choices in a very understandable way. Relevant figures are provided for context of the architecture and tasks. The experiments are well motivated and help understand the performance of PIDE / GPIDE across tasks, RL methods, and transfer scenarios.


#### Significance

GPIDE seems broadly applicable across deep RL control tasks, especially those related to tracking and navigation / locomotion as demonstrated by experiments. The method is an approach to handling the POMDP problem on control problems as a drop-in architecture for history encoding.


**Weaknesses:**

#### Experiments
Compared to RNNs such as GRU which consider only the current timestep and state to predict the next action, GPIDE seems to consider all past timesteps from 1 to t within each step, as part of the weighted summation resembling the PID I step, which may be more expensive than other methods if the time / observations are large. It may be worthwhile to consider potential inference time compute differences because of this, and also to consider ablations or baselines which also have such history context.

#### Significance
On line 116-117, the authors mention that LSTM, GRU, and transformers were shown to be powerful tools for NLP because of complex relationships within the task. Because control tasks may not have such complexities, such methods may overfit rather than help, especially in cases of domain transfer. However, these methods have also been applied to success in domains including robotics and visual encoding (e.g. https://arxiv.org/abs/2212.06817, https://arxiv.org/abs/2010.11929, https://arxiv.org/abs/2106.01345) - perhaps this indicates that the best targets for using GPIDE would be cases where highly-overparameterized / “large” models may overfit, which would be towards the simpler side.


**Questions:**

#### Limitations
The authors mention that control tasks which require memory may not be suitable - it would be further interesting to consider from the perspective of task similarity to that of PID control. Do the authors think that there would be correlation between similarity to PID / tracking tasks and successful application of GPIDE?

For instance, a common image-based task would be visual grasping which involves reasoning about object geometry, where it may be less clear that results would directly transfer.

Furthermore, some tasks may be less partially observable in nature than others - given this method focuses on a history encoder, would the relevance of history generally for certain tasks be relevant to consider?


**Limitations:**

The limitations are generally adequate (there are some questions in the section above).

---

> ### Author Rebuttal · Authors · 2023-08-09
>
> Thank you for your time and your thorough review! We were happy to see that you thought the paper was high-quality and that the method seems broadly applicable across RL tasks. We have touched on many of your points in the weakness section in the global response (particularly computational expense, ablation of history context, and extension to image-based tasks). You mentioned that perhaps the best targets for using GPIDE are cases in which the tasks or dynamics are relatively simple. This may very well be the case; however, we would like to point out that the papers you linked use a supervised loss function during training. Besides a few works we mention in the paper, it seems that most successes applying transformers to control use supervised losses. We found that training transformers in an online RL setting where there are bootstrapped target values is far more challenging. As such, it is not clear to us exactly when a transformer should be used over GPIDE for online RL.

---

### Official Review · Reviewer_EZtX · 2023-07-25

**Soundness:** 3 good
**Presentation:** 3 good
**Contribution:** 2 fair
**Rating:** 5
**Confidence:** 4

**Summary:**

This paper proposes a new way to encode features in partially observable environments using PID. Experiments show superior results over several domains compared with recurrent and transformer encoders.

**Strengths:**

The paper is easy to read overall.

Experiments show promising performance on multiple domains.

**Weaknesses:**

Some assumptions and technical details need to be clarified.

1) Using the difference between observations as the derivative of the error (D) is trivial,  assuming that the reference value for the observation at different timesteps is the same, which is not true.

2) The author mentioned in Line 102 that using M separate PID controllers to control M signals has some clear shortcomings. This paper models the MIMO problem in a centralized manner, inputting all inputs and outputting a 3M dimensional vector. However, this would not be feasible if there is a large scale of signals that need to control.

3) It said the $f_{\theta}^h$ is a learnable linear projection, how to learn it is not discussed. Also with $g_{\theta}$.

4) How to determine the suitable length of the history, As for the length $1:t$ in $v_{1:t}^{h}$?

More ablations should be provided.

1) The ablations investigate the influence of different types of heads, while why choosing these three settings (ES, ES+Sum, Attention) is not explained. What about ES+Attention? Why five ES + 1 Sum? Are there any insights in choosing the number and combination of different heads?

2) The influence of different lengths of history should also be investigated.

More limitations should be discussed. For example, the encoder processing time is highly dependent on the length of the history.

Some related works are not discussed in this paper. For example, [1-3].

[1] An adaptive deep reinforcement learning approach for MIMO PID control of mobile robots

[2] Self-Tuning Two Degree-of-Freedom Proportional–Integral Control System Based on Reinforcement Learning for a Multiple-Input Multiple-Output Industrial Process That Suffers from Spatial Input Coupling

[3] Robust Adaptive PID Control based on Reinforcement Learning for MIMO Nonlinear Six-joint Manipulator




**Questions:**

Questions,

Please see the pros and cons part about the assumptions, technical details, ablations, and limitations.



**Limitations:**

Please see the pros and cons part about the limitations.

---

> ### Author Rebuttal · Authors · 2023-08-09
>
> Thank you for your time and your review! We will respond to each of your weaknesses in a corresponding list. Starting with the list of assumptions and technical details,
>
> 1. For your first point, you are correct that the reference value need not be the same throughout time. However, GPIDE can still recover the correct D term. To demonstrate, consider an M=1 dimensional tracking problem. As per line 89, $o_t = (x_t, \sigma_t)$. Then, $o_t - o_{t - 1} = (x_t - x_{t-1}, \sigma_t - \sigma_{t - 1})$. If we let our $f$ be a projection that simply subtracts the second coordinate from the first we get $f(o_t - o_{t - 1}) = (x_t - \sigma_t) - (x_{t - 1} - \sigma_{t - 1})$. Thus, we have recovered the correct D term up to a constant.
>
> 2. While we agree that learning a policy becomes difficult as the dimensionality $M$ grows, we assert that all approaches will struggle as $M$ becomes large, especially since policies need to remember a history of observations to perform well in the POMDP setting. For example, a GRU-based approach would learn a $D$-dimensional encoding of the history, which effectively makes the policy over a $(M + D)$-dimensional input. We found that $D$ needs to be relatively large for good performance (64 for tracking experiments and 256 for locomotion experiments), and it would likely need to grow as $M$ grows. Moreover, the mapping from the history to the $D$-dimensional encoding is often learned simultaneously with the policy. Adding up all these factors, we assert that learning a policy that takes in a fixed, $3M$-dimensional input is actually a less daunting task than using a GRU or LSTM.
>
> 3. $f$ and $g$ are simply fully connected layers with no activation functions. Everything is trained end-to-end using the Adam optimizer. We will add a mention of this in the final paper.
>
> 4. For the lookback $t$, we simply use the full episode for tracking problems and the same setting as was used by the baseline methods in Ni et al. In practice $t$ is a hyperparameter; however, this is not unique to our method. Although recurrent networks can encode arbitrary history lengths at test time, one must often select how far to look back during training the recurrent network. Please refer to the global response for more discussion on this.
>
> For your comment on the ablations:
>
> 1. We chose the ablations we did because we wanted to investigate two questions: how important are heads that accumulate information (hence ES vs ES + Sum)? and do we even need ES if attention has the capacity to represent it already (hence ES vs Attention)? This reasoning was not presented well in the paper, and our final version will flesh this out. While the results of ES + Attention would be interesting to see, we did not feel the need to devote computational resources to this configuration since it would not answer either of our two questions. As to why we picked 5 ES heads and 1 summation head, ES can capture several different time scales depending on $\alpha$ whereas summation can only capture one. Therefore, it seemed logical to only have one summation head.
>
> 2. Please refer to the global comment for time horizon ablations.
>
> Thank you for bringing these related works to our attention. We will cite them in our final paper. While they cover how RL can improve PID rather than how PID can improve RL (like our paper), we still think it is important related work to discuss.
>
> If you feel that we have adequately addressed your concerns, we would respectfully ask you to consider raising your score. Thank you!

---

### Official Review · Reviewer_vEC9 · 2023-07-27

**Soundness:** 3 good
**Presentation:** 3 good
**Contribution:** 2 fair
**Rating:** 5
**Confidence:** 4

**Summary:**

This paper studies the problem of learning from histories in partially observable MDPs where a key question is how to design an architecture that is general enough so as to work on a large set of problems, yet specific enough to be sample efficient.

Inspired by the success of PID in the more classical control literature, the idea here is to design an architecture that is solely comprised of summation, attention, and linear operations.

Experiments show that on tracking problems, where PID is originally proposed for, as well as more complicated control problems, this simple architecture is good enough to achieve competitive results.

**Strengths:**

The paper shows that using a complicated architecture is an overkill for many of the domains that the RL community considers as benchmarks. On this note I'd like to mention that, while the contexts are somewhat different, a similar conclusion can be made from "Towards Generalization and Simplicity in Continuous Control" Rajeswaran et al, where they show that simpler architectures such as linear models are equipped to deal with Mujoco Tasks. The difference is that in their paper they were considering reward-maximization in presence of the full state, whereas here the goal is learn a memory-like function in conjunction with performing well in the given task.

In any case, I generally like results that show simple stuff work, because simple things are easy to understand, implement, and execute.

**Weaknesses:**

Overall I like this paper but I also have some concerns:

Fundamentally, I feel like the results in this paper are pertinent in terms of a bigger question, namely how rich we want our hypothesis class to be for learning a certain target. The solution adopted by the ML community nowadays seems to be: we really do not know how complicated a task is, so by choosing a very complicated architecture and the fact that we are at the mercy of universal function approximation capability, we can always in principle find the right fit. Moreover, the double descent phenomenon  ("Deep double descent: where bigger models and more data hurt", Nakkiran et al) tells us that empirically we are very likely to find a good fit with larger and more complicated models and that, despite more classical belief, such an overparameterized fit happens to also be quite robust. ("A Universal Law of Robustness via Isoperimetry" Bubeck and Sellke)

Now, moving to the RL setting, these insights from supervised learning do not really carry over because we usually learn by bootstrapping off of Bellman targets, and by increasing the complexity of our function approximator, we also have to learn from increasingly more complicated targets (unlike supervised learning where targets are fixed).

So I think, at least with our current RL algorithms, unlike SL we always need to choose our function class carefully and so adding inductive bias, such as ones provided in this paper, can usually be helpful. My worry is that adding such biases are helpful in a limited sense: It is not clear how much inductive bias should I add for a given task when I don't know too much about that task. Sure, I can always choose environments and problem settings where I know that my inductive bias is conducive to solving the problem, but do we really want to bring back the burden to ML practitioners to carefully think about the kind of inductive biases that are useful for their task?


Grounding my concerns more so in the experimental results provided here, for example we see that the inductive bias provided is already giving small (or negative) gains as we move to some of the more complicated domains such as HalfCheetah and Ant (based on Figures 18 and 19 in the Appendix), so I ask authors, are you not worried that as we move to much larger domains then it becomes increasingly more difficult to show performance gains for GPIDE compared to other approaches? In other words, does the inductive bias scale reasonably with problem complexity or there is only so much this could be useful?

To further play devil's advocate, to say that GPIDE is capable of beating the transformer architecture is sort of a weird claim, because as authors say GPIDE could be thought of as a special case of transformers, and further as you show the attention layer is often not very useful. This tells me that with proper hyper-parameter tuning and regularization, the transformer architecture should in principle be able to always do better than GPIDE, and we can still hope to find the transformer useful as we move to larger problems.

**Questions:**

I would like to ideally see an acknowledgment that as we move to much more difficult domains, GPIDE may be incapable of doing well simply because the actual function to be represented in not in the function class. But such an example is currently absent. This may give a false sense that the paper is claiming that GPIDE is a general-purpose memory learner but that is not true, and I am not saying that authors say it is true, but having examples of failure cases would really clarify this point.

If I may say, I would also like to see experiments on different types of problems. For example, can we expect GPIDE to perform reasonably on image-based domains? Like the paper mentions, we can hope to apply the GPIDE on the embedding space learned by some more capable function approximator, but it seems to me that to learn good embeddings that are useful for downstream memory learning, one needs to leverage some kind of GRU/Transformer/LSTM architecture, and so it is not immediately clear to me how to leverage GPIDE in this setting.

My other question is also that as we know RNNs struggle to learn important events from the distant past because the vanishing/exploding gradients. LSTMs and Transformers are less prone to this issue but I wonder if we could better compare this capability of GPIDE in contrast to the existing architectures. In the Half-Cheetah experiment, for example, do we have a good sense of how far back one needs to go to be able to accurately learn the position and/or the velocity signal?

Also, from the intro the paper reads "Another hurdle stems from the fact that policies are often trained in an imperfect simulator, which is likely different from the true environment. Combining these two challenges necessitates striking a balance between extracting useful information from the history and avoiding overfitting to modelling error."

My thoughts reading this sentence was that we are going to either have actual sim2real experiments or some kind of a transfer learning benchmark, where we learn the memory function in some settings, and test the learned memory function in another test setting. This would have been more convincing in terms of claiming that GPIDE is not prone to overfitting. Did I get it right that this experiment is absent?

---

> ### Author Rebuttal · Authors · 2023-08-09
>
> Thank you for your time and your thoughtful review! We greatly appreciate the feedback, and we are glad that you liked the paper.
>
> * Your comments highlight a fascinating problem which is the tradeoff between the benefits of high capacity modeling and the benefits of useful inductive biases. As you point out, the community is getting a better handle on these tradeoffs for supervised learning, but for RL many of these questions are fairly open. We agree that our Limitation section should better acknowledge GPIDE’s reduced capacity being a potential problem for high dimensional or complex environments, and we will expand our discussion in the final version of the paper.
>
>    While a quick look at the margin of victory might suggest we do better on lower dimensional environments, we note that GPIDE consistently improves upon or matches the performance of the best competitor (see global response for more information on HalfCheetah-V). It is important to note this best competitor is not always the same. In particular, while SAC-LSTM is the best performing competitor by a substantial margin on Ant-P, it is also the worst performing competitor on Ant-V. We believe this robustness across environments is encouraging for many other problems including larger and more complex ones.
>
>    Regarding the possibility of transformers always doing better than GPIDE, it is not clear to us that this is true with the right hyper-parameters and regularization. We view this as analogous to CNNs vs fully connected networks (FCNs) on image tasks. While the FCNs are a more expressive class, they are rarely as performant as their CNN counterparts, which utilize an intelligent inductive bias. As you already pointed out, there are still many open questions around RL and whether these same phenomena apply.
>
> * We have done an additional experiment on the amount of lookback for HalfCheetah. Please refer to the global response.
>
> * We do have substantial experiments on sim2real and the generalization capabilities of PIDE and GPIDE. For the MSD and DMSD environments, we train the policy on a reduced set of system parameters and test on a larger set. This is the same type of experimental setup used in “Assessing generalization in deep reinforcement learning” (Packer et al 2018). We expand on this in the navigation experiments by having the “sim” not model friction whereas the “real” environment has friction. Perhaps most realistic, in our fusion experiment we create a simulator using first-principle equations. For the “real” environment we use a data-driven dynamics model trained using data from the actual device. In all of these experiments, PIDE and GPIDE showed an impressive level of generalization, especially when compared to using a GRU.

---

> > ### Comment · Reviewer_vEC9 · 2023-08-11
> >
> > Awesome, thanks for the rebuttal.
> >
> > As alluded to by a few other reviewers (64AN RWrF), we still have a lingering question about scaling up GPIDE to larger domains, or domains with different inputs such as text or image.
> >
> > That said, I am really encouraged by your acknowledging that in the limitation section a more nuanced discussion on the potential limitation of this inductive bias would be highly appropriate. This would hedge against potentially over-claiming the result. This seems to be the kind of inductive bias that can a) work surprisingly well in lower dimensional settings and b) provide enough insights for follow up works to propose more scalable inductive biases. So overall, I keep my score and remain supportive of this paper getting accepted.

---

> > > ### Author Response · Authors · 2023-08-15
> > >
> > > Thank you for your reply and for your support for our paper's acceptance! We sincerely value your feedback.

---

### Author Rebuttal · Authors · 2023-08-09

Thank you to all of the reviewers for taking the time to read our paper and give thoughtful reviews. We value your feedback and hope to use it to strengthen our work. In this global comment we will address some points that were raised in multiple reviews.

### Performance of HalfCheetah-V

Some reviewers pointed out that GPIDE achieves substantially worse performance on HalfCheetah-V and were concerned this may indicate the GPIDE architecture does not have enough flexibility to handle higher dimensional tasks. However, we assert this is not due to the GPIDE architecture but due to attention heads. Looking at Table 28 in the Appendix, we see that removing attention heads more than doubles the average performance ($20.39 \pm 29.60$ to $53.14 \pm 5.86$), making it competitive with the TD3-GRU, which is the best performing method ($59.03 \pm 2.88$). Moreover, GPIDE-Attention is one of the worst performers, showing attention heads result in particularly poor performance in this environment. We agree that the current iteration of the paper is lacking this explanation, and we will update our paper with these details.

### Computational Cost of GPIDE

Many reviewers inquired about the computational cost of GPIDE. This information can be found in Appendix E, and the final version of the paper will do a better job of referencing this in the main body. Using attention is slower since the current query must be compared with all previous keys in order to form the encoding. We see in practice this results in a roughly 20% slow down on the tracking problems. However, we would like to emphasize that if no attention heads are used (which ablations suggest is optimal), $w_{t-1}$ can be cached at every time step and only $v_t$ need be computed to calculate $w_t$. Since $v_t$ is the result of a linear projection, this would likely be faster than using a GRU or LSTM. Unfortunately, our implementation of GPIDE-ES and GPIDE-ESS does not take this shortcut into consideration since our code was created with flexibility of head type in mind.

### Impact of Different Lookback Lengths

Multiple reviews wished to see the impact of lookback length on the policy performance. We were able to run preliminary experiments with varying lookback on the HalfCheetah environments for GPIDE (see one page pdf), and will run similar experiments for all locomotion environments and variations of GPIDE found in the ablations for the full paper. As a reminder, we picked a lookback of 64 since this is the sequence length that competitor algorithms trained on; however, we agree that investigating the lookback is important to understanding GPIDE.

For HalfCheetah-P, it is clear that a greater amount of lookback corresponds to higher returns. In fact, it seems that we are able to get even stronger performance by increasing the history from 64 to 128.

For HalfCheetah-V, there is a clear increase in performance going from 4 to 16 lookback; however, performance drops off after this and crashes for a lookback of 128. As stated in the global response addressing HalfCheetah-V, we believe attention heads are the cause for this, and it seems that there are instabilities that occur as lookback grows. We do not expect to see these same instabilities when we run the same experiments for GPIDE-ES and GPIDE-ESS. As initial evidence to this claim, we also have preliminary data for GPIDE-ESS with a lookback of 128. Although these jobs are still running, it is clear that the stability problems have been eliminated, and it seems the final performance will be greater than any of the other GPIDE variants.

### Extension to Image-Based Tasks
Many reviewers also mentioned the extension to image-based tasks. While we are optimistic about the application of GPIDE on such tasks given the right image encoding, working with images comes with its own set of challenges which we leave for future work.

---

### Decision · Program_Chairs · 2023-09-21

**Decision:**

Accept (poster)

**Comment:**

This paper addresses the problem of encoding an RL agent's history to provide a state representation in partially observable RL tasks. The proposed architecture is a feedforward encoder that learns multiple "heads" which each encode past history in a different way. The architecture is evaluated in RL tasks, mainly focusing on tracking but also partially observable locomotion tasks

Main strengths: The architecture is novel to the best of reviewer knowledge. Seems like an interesting in-between architecture between recurrent architectures and the feedforward approach of simply concatenating history. Empirical evaluation shows improved policy learning in a range of partially observable learning tasks.

Main Concerns: I find the connection to PID control tenuous and would encourage the authors to clarify how the novel architecture is PID-like or to downplay the PID connection. It was hard to see the connection to PID control. I think the empirical study quality could be improved with additional trials ran for experiments to provide more confidence in results. I downplay this concern some because of the large number of tasks considered. A nice improvement could be for the authors to consider using the tools provided by the author of this paper: https://arxiv.org/pdf/2108.13264.pdf.

History encodings for handling partially observable tasks are a useful contribution for RL and the proposed architecture seems useful for this. I would recommend acceptance.